EMBO
Molecular Medicine

# Follistatin-like 1 promotes cardiac fibroblast activation and protects the heart from rupture

Sonomi Maruyama[1], Kazuto Nakamura[1], Kyriakos N Papanicolaou[1], Soichi Sano[1], Ippei Shimizu[1], Yasuhide Asaumi[1], Maurice J van den Hoff[2], Noriyuki Ouchi[3], Fabio A Recchia[4,5] & Kenneth Walsh[1,*]

## Abstract

Follistatin-like 1 (Fstl1) is a secreted protein that is acutely induced in heart following myocardial infarction (MI). In this study, we investigated cell type-specific regulation of Fstl1 and its function in a murine model of MI. Fstl1 was robustly expressed in fibroblasts and myofibroblasts in the infarcted area compared to cardiac myocytes. The conditional ablation of Fstl1 in S100a4-expressing fibroblast lineage cells (Fstl1-cfKO mice) led to a reduction in injury-induced Fstl1 expression and increased mortality due to cardiac rupture during the acute phase. Cardiac rupture was associated with a diminished number of myofibroblasts and decreased expression of extracellular matrix proteins. The infarcts of Fstl1-cfKO mice displayed weaker birefringence, indicative of thin and loosely packed collagen. Mechanistically, the migratory and proliferative capabilities of cardiac fibroblasts were attenuated by endogenous Fstl1 ablation. The activation of cardiac fibroblasts by Fstl1 was mediated by ERK1/2 but not Smad2/3 signaling. This study reveals that Fstl1 is essential for the acute repair of the infarcted myocardium and that stimulation of early fibroblast activation is a novel function of Fstl1.

**Keywords** cardiokine; fibrosis; infarct healing; myocardial infarction
**Subject Category** Cardiovascular System

## Introduction

The drive to identify new biomarkers and therapeutic targets for heart disease has led to increased interest in understanding the regulation and function of the cardiac secretome. Factors secreted from the heart, referred to as "cardiokines", have been shown to be involved in the homeostatic control of cardiac function, modulating the response to injury and communicating with remote organs and tissues (Walsh, 2009; Shimano *et al*, 2012). Follistatin-like 1 (Fstl1),

also known as TSC-36, is a secreted glycoprotein that belongs to the follistatin and SPARC (secreted protein, acidic and rich in cysteine) families (Hambrock *et al*, 2004). Fstl1 was initially identified as transforming growth factor-beta 1 (TGF-β1)-induced protein in a murine osteoblastic cell line (Shibanuma *et al*, 1993). In the heart, Fstl1 is a cardiokine whose levels are upregulated in models of acute and chronic injury, including MI, pressure overload-induced hypertrophy, and ischemia/reperfusion (I/R) injury (Oshima *et al*, 2008). In clinical studies, serum Fstl1 levels are significantly higher in patients with acute coronary syndrome (ACS) (Widera *et al*, 2009, 2012) and chronic systolic heart failure (El-Armouche *et al*, 2011).

The overexpression of Fstl1 has been shown to be protective in a number of models of myocardial injury. In a murine model of myocardial I/R injury, adenovirus-mediated Fstl1 overexpression reduced infarct size and diminished cardiac myocyte apoptosis (Oshima *et al*, 2008). Similarly, in a porcine model of myocardial I/R, the delivery of recombinant Fstl1 protein diminished scar size likely due to its anti-apoptotic and anti-inflammatory actions (Ogura *et al*, 2012). More recently, it has been shown that the reconstitution of a non-glycosylated isoform of Fstl1, produced by the epicardium, promotes the proliferation of immature cardiac myocytes and diminishes infarct size post-MI (Wei *et al*, 2015). In contrast, the effects of Fstl1 loss of function in heart have only been evaluated in a model of pressure overload-induced hypertrophy (Shimano *et al*, 2011). In this model, the genetic, cardiomyocyte-specific ablation of Fstl1 led to an exaggerated hypertrophic response and facilitated the transition to systolic dysfunction following transverse aortic constriction (Shimano *et al*, 2011). Collectively, these data suggest that the injury-induced upregulation of Fstl1 plays a clinically relevant role in the modulation of myocardial pathological processes.

Permanent left anterior descending (LAD) artery ligation in mice leads to exceptionally high levels of Fstl1 induction that far exceed what is observed in other models of cardiac injury (Oshima *et al*, 2008). However, the functional role of endogenous Fstl1 in experimental MI has not been explored previously using loss-of-function genetic models. In this study, we tested the hypothesis that cardiac fibroblasts are the major source of injury-induced Fstl1 expression following experimental MI. The partial ablation of

1 Department of Molecular Cardiology, Whitaker Cardiovascular Institute, Boston University School of Medicine, Boston, MA, USA
2 Department of Anatomy, Embryology and Physiology, Academic Medical Center, Amsterdam, The Netherlands
3 Molecular Cardiovascular Medicine, Nagoya University Graduate School of Medicine, Showa-ku, Nagoya, Japan
4 Cardiovascular Research Center, Lewis Katz School of Medicine at Temple University, Philadelphia, PA, USA
5 Institute of Life Sciences, Scuola Superiore Sant'Anna, Pisa, Italy
*Corresponding author. Tel: +1 617 414 2390; Fax: +1 617 414 2391; E-mail: kxwalsh@bu.edu

fibroblast-encoded Fstl1 led to perturbations in the fibrotic response to LAD ligation and caused an increase in mortality due to cardiac rupture. These data reveal a novel role for Fstl1 in cardiac repair.

## Results

### Fstl1 upregulation after MI

To assess the time-dependent change of Fstl1 induction after MI, we examined Fstl1 protein expression in left ventricle and serum of C57BL/6 WT mice at multiple time points (Fig 1A). Fstl1 upregulation was detected in both ischemic and non-ischemic (remote) portions of the myocardium at 1 day post-MI. Expression progressively increased at subsequent days and appeared maximal at 7 days post-MI in the experimental time course. Elevated expression was still detected at 28 days post-MI in both regions of the myocardium. Throughout the time course, the induction of Fstl1 was most robust in the ischemic area versus the remote region of the left ventricle. For example, at the 1 day time point, Fstl1 was induced above baseline by 16-fold in the ischemic zone and 2.2-fold in the remote zone. Circulating Fstl1 was also detected in serum by immunoblot analysis, and the timing of induction was similar to that observed in the myocardium. At day 7 post-MI, serum Fstl1 displayed the greatest induction (three-fold) in the experimental time course (Fig 1B). Widera et al reported a correlation between serum Fstl1 level and the severity of myocardial damage in ACS in the patient population (Widera et al, 2009, 2012). Correspondingly, we found a significant correlation between myocardial damage, measured by cardiac myosin light chain-1 in the serum, and both of myocardial and serum levels of Fstl1 (Appendix Fig S1A and B).

### Fstl1 is predominantly expressed in non-cardiomyocyte compartment of the infarcted heart

The level of Fstl1 induction post-MI was markedly greater than what was previously observed in a model of pressure-overload hypertrophy where the predominant source of Fstl1 was determined to be cardiac myocytes (Shimano et al, 2011). Thus, we assessed the identity of the Fstl1-producing cells in infarcted heart by immunohistochemistry. Fstl1-positive cells were detected mainly in the interstitium of left ventricle (Fig 2A). Intensely positive Fslt1-expressing cells were more abundant in the ischemic area than in the remote, non-ischemic area of the heart (Fig 2A). At this level of resolution, immunofluorescent Fstl1 staining did not appear to colocalize with sarcomeric actinin-positive cells in the infarcted heart suggesting that cardiomyocytes are not a major source of Fstl1 under these conditions (Fig 2B).

S100a4, also known as fibroblast specific protein 1 (Fsp1), has been used as a fibroblast marker (Strutz et al, 1995; Kovacic et al, 2012). Immunofluorescence staining revealed that a subset of S100a4-positive cells were also positive for Fstl1 expression (Fig 2C). Partial colocalization of Fstl1 with vimentin-positive cells was also detected (Appendix Fig S2A). In contrast, Fstl1 did not colocalize with CD-31 or lectin BS-1, markers of vascular endothelial cells (Appendix Fig S2B), Mac2 or CD68, macrophage markers (Appendix Fig S2C), and MPO, a neutrophil marker (Appendix Fig

S2D). It has been reported that endothelial cells, smooth muscle cells, and CD68-positive macrophages all express S100a4 to some degree (Schneider et al, 2007). However, in this study, we found neither macrophage nor endothelial cell markers to be positive for Fstl1 (Appendix Fig S2B and C).

Following myocardial infarction (MI), fibroblasts transdifferentiate to myofibroblasts that are α-SMA positive and play a critical role in extracellular matrix synthesis and lesion repair (Chen & Frangogiannis, 2013). Fstl1 and α-SMA double-positive cells could be detected in interstitium of infarcted area (Fig 2D), suggesting that myofibroblasts also express Fstl1 in infarcted heart. There was a significant correlation between myocardial levels of Fstl1 and α-SMA protein expression in mice after MI (Appendix Fig S3). Consistently, S100a4-positive cells partially colocalized with α-SMA-positive myofibroblasts in these sections (Appendix Fig S2E).

### Attenuated Fstl1 protein induction after MI in S100a4$^{cre+/-}$ × Fstl1$^{flox/flox}$ mice

S100a4 mRNA is upregulated by more than 50-fold in the ischemic area of WT mice at 7 days after MI (Fig 3A). Thus, to examine the function of Fstl1 in this cell population, S100a4$^{cre+/-}$ × Fstl1$^{flox/flox}$ mice (Fstl1-cfKO) were generated. Fstl1-cfKO and littermate S100a4$^{cre-/-}$ × Fstl1$^{flox/flox}$ mice (control, WT mice) underwent permanent LAD ligation procedure to generate MI. The MI-induced Fstl1 induction in the ischemic lesion was significantly suppressed by ~40% at the transcript level and 70% at the protein level in Fstl1-cfKO mice compared to WT mice (Fig 3B and C). Tgfβ1 mRNA that encodes an upstream modulator of Fstl1 did not differ between WT and Fstl1-cfKO under sham or infarcted conditions (Fig 3B). Loss of Fstl1 was also examined in cardiac fibroblasts isolated from Fstl1-cfKO and WT neonatal mice. Fstl1 protein was reduced ~50% in cardiac fibroblasts (CFs) from Fstl1-cfKO mice compared to WT mice CFs ($P = 0.0013$, unpaired $t$-test, two-tailed, Fig 3D). Secreted Fstl1 protein from CFs was also significantly diminished in Fstl1-cfKO mice CFs ($P = 0.00003$, unpaired $t$-test, two-tailed, Fig 3D). Collectively, these data document a partial but significant reduction in the MI-induced levels of Fstl1 in the whole heart, and in the cardiac fibroblast population, of S100a4$^{cre+/-}$ × Fstl1$^{flox/flox}$.

Previous work has reported a modest level of Fstl1 expression by cardiac myocytes (Shimano et al, 2011) and expression of Fstl1 by activated macrophages (Miyamae et al, 2006; Chaly et al, 2014). In the infarct model that leads to very high levels of Fstl1 induction (Oshima et al, 2008), co-immunolocalization did not detect appreciable Fstl1 expression by either of these cell types in the myocardial infarct region (Fig 2 and Appendix Fig S2C). Regardless, additional genetic experiments were undertaken to test the possibility that Fstl1 expression by cardiomyocytes or macrophages contributes appreciably to the overall upregulation of Fstl1 in the heart post-MI. Although cardiomyocyte-specific Fst1 knockout mice (αMHC$^{cre+/-}$ × Fstl1$^{flox/flox}$) showed a significant decrease (~50%) of Fstl1 in the TAC model (Shimano et al, 2011), there was no appreciable attenuation of Fstl1 protein overexpression in MI model (Fig EV1A). To evaluate the participation of myeloid-derived cells/macrophages in MI-induced Fstl1 production, LyzM$^{cre+/-}$ × Fstl1$^{flox/flox}$ mice were created and subjected to LAD ligation. Similar to cardiomyocyte-ablated Fstl1 mice, Fstl1 ablation in myeloid-derived cells did not show reduction of Fstl1 induction

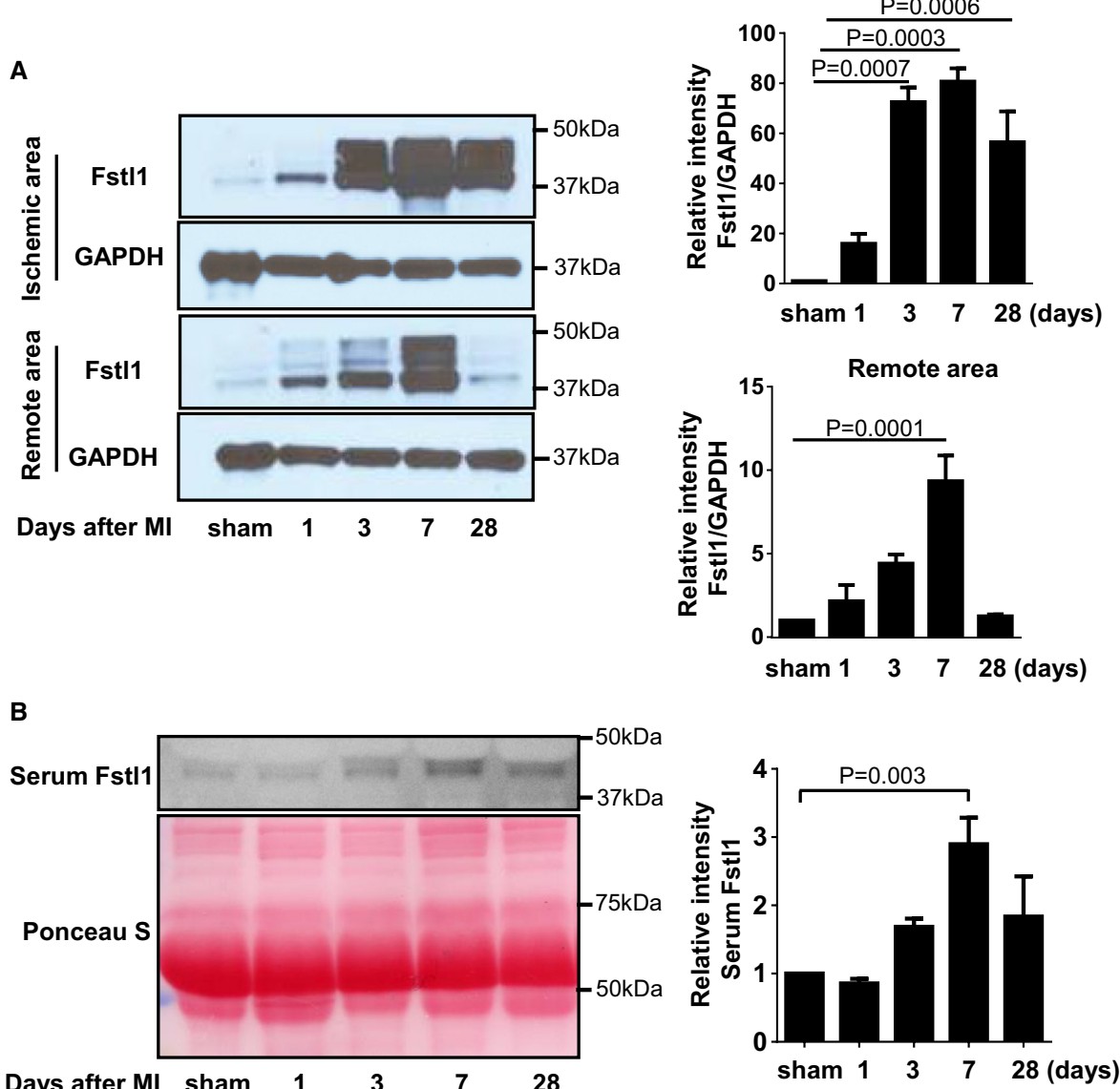

**Figure 1. Fstl1 expression is induced in ischemic heart.**

A   Detection of Fstl1 protein in ischemic myocardium of WT mice. After MI induction, the heart was harvested at the indicated time points and separated to ischemic and remote (non-ischemic) areas. Fstl1 protein was detected by Western blotting ($n$ = 3, each time point).

B   Serum Fstl1 was detected by Western blotting at the indicated time points. Ponceau S staining of serum reveals equivalent amount of loaded protein ($n$ = 4, each time point).

Data information: Error bars represent mean ± SEM. Statistical analysis was performed by ordinary one-way ANOVA. *Post hoc* test was performed by Dunnett's test. Source data are available online for this figure.

after MI (Fig EV1B). Collectively, these results indicate that non-cardiomyocytes/non-myeloid, mainly cardiac fibroblasts and myofibroblasts, are a principal source of Fstl1 protein induction after MI.

**Fstl1 protects the post-MI heart from rupture**

Fstl1-cfKO and WT mice underwent permanent LAD ligation surgery, and survival rates were recorded (Fig 4A). Mortality after

MI was significantly higher in Fstl1-cfKO mice compared to WT mice (log-rank (Mantel–Cox) test *P* = 0.031, hazard ratio (Mantel–Haenszel) = 2.482 (cfKO/WT), 95% CI of ratio = 1.088 to 5.658, Fig 4A and Table 1). The mortality after MI in control WT mice was consistent with the frequencies reported previously by van den Borne *et al* (2009). All mice died due to cardiac rupture, as indicated by the presence of a hemothorax upon autopsy (Fig 4B), with the exception of one Fstl1-cfKO mouse that died of unknown causes. Specifically, the rate of mortality was 46.7% for Fstl1-cfKO group (14

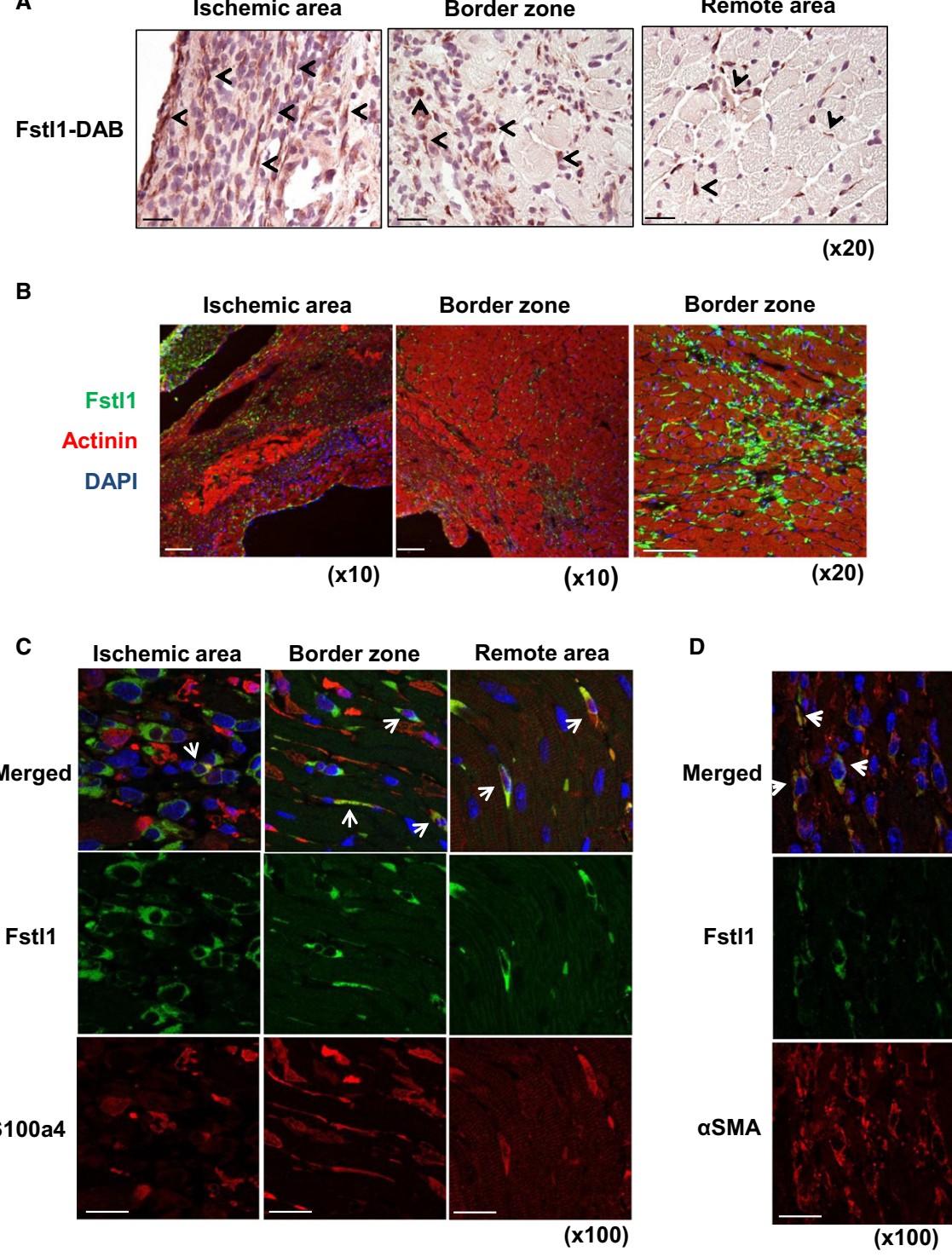

**Figure 2. Fstl1 protein expression is induced in non-cardiomyocyte cells after left anterior descending (LAD) ligation.**

A   DAB immunohistochemical staining of Fstl1 in ischemic heart at day 3 after LAD ligation. Arrows indicate a subset of Fstl1-positive cells. Counter staining was performed with hematoxylin. Scale bar indicates 50 μm.

B   Immunofluorescent staining of Fstl1 (green) and sarcomeric actinin (red). Fstl1 protein was detected in interstitial areas of the ischemic and border zone but not in cardiomyocytes. Scale bar indicates 100 μm.

C   Immunofluorescent staining of Fstl1 (green) and S100a4 (red) in the ischemic area. Arrows indicate Fstl1 and S100a4 double-positive cells. Scale bar indicates 20 μm.

D   Immunofluorescent staining of Fstl1 (green) and α-smooth muscle actin (red). Arrows indicate Fstl1 and α-SMA double-positive cells. Scale bar indicates 20 μm. Nuclei were stained by DAPI.

deaths per 30 mice) and 27.1% for WT group (16 deaths per 59 mice) within 7 days of MI surgery. All WT and Fstl1-cfKO mice underwent sham surgery survived ($n = 16$ and $n = 15$, respectively). The following number of survived mice were euthanized at 7 days after the surgery for analysis ($n = 15$ for WT MI, $n = 14$ for Fstl1-cfKO MI). When mice were euthanized for organ sampling, one Fstl1-cfKO mouse revealed an impending left free wall rupture as indicated by the existence of a hematoma in the left ventricular wall (Fig 4B). Histological analysis of necropsy specimens from Fstl1-cfKO mice revealed the existence of ruptures in the endocardium and intrawall hematomas in the left ventricular free wall (Fig 4C). Echocardiographic analysis was performed on sham treated WT and cfKO mice ($n = 16$ and 14, respectively) and on WT and cfKO mice surviving mice at 6 days post-MI in ($n = 15$ and 14, respectively). Of the surviving mice, echocardiographic analyses did not reveal a statistically significant difference in cardiac function, nor was there a difference between WT and Fstl1-cfKO mice under sham conditions (Table 2). Likewise, there were no differences in blood pressure between the experimental groups (Table 2).

At 4 weeks post-MI, surviving cfKO mice displayed a significant reduction in fractional shortening compared to WT mice (Table EV4). While there was a trend toward greater infarct size in the cfKO mice at this time point, this difference was not statistically significant (Appendix Fig S4).

### Fstl1 deficiency alters matrix remodeling in post-MI heart

The upregulation of extracellular matrix synthesis in fibroblasts and myofibroblasts, such as fibrillary collagen (collagen type I and III) and fibronectin after MI, is essential for tissue repair (Squires *et al*, 2005). The expression of fibrotic gene transcripts *Col1a1* and *Fn1* were strongly upregulated in ischemic lesion of WT mice heart. There were trends toward reduced *Col1a1* and *Fn1* mRNA expression in ischemic lesions of Fstl1-cfKO mouse hearts ($P = 0.104$ and $P = 0.263$ for *Col1a1* and *Fn1*, respectively by Tukey's test), $n = 15$ for WT-IA and $n = 13$ for cfKO-IA) in these animals (Fig 5A). The induction of collagen I and fibronectin protein expression in ischemic tissue was significantly attenuated in Fstl1-cfKO compared to WT mice in the ischemic heart tissue (Fig 5B).

α-SMA is a marker of myofibroblasts as well as vascular smooth muscle cells. As expected, α-SMA protein levels were strongly upregulated in ischemic lesions of WT mice myocardium. However, the induction of α-SMA protein expression was significantly attenuated in the infarcted Fstl1-cfKO mouse hearts (Fig 5B). To evaluate the extent of myofibroblast composition in the infarct region, α-SMA immunohistochemical staining was performed. Markedly fewer α-SMA-positive myofibroblasts were observed in Fstl1-cfKO mice heart compared to WT mice heart after MI (Fig 5C).

These results suggest that myofibroblast activation or differentiation is attenuated in Fstl1-cfKO mouse hearts after MI, likely contributing to the reduction in fibrotic gene expression. Polarized light microscopy was used to assess collagen fiber architecture in the infarcted regions of the WT and Fstl1-cfKO mice at 7 days post-MI. The birefringence of collagen was predominantly orange-red in WT mice, but green-yellow in the Fstl1-cfKO (Fig 5D). These birefringence properties are consistent with a more tightly packed collagen lattice in the WT mice, but a loosely assembled collagen structure in the Fstl1-cfKO infarct zone.

It has been reported that Fstl1 has a pro-inflammatory activity and that its serum levels are elevated in juvenile rheumatoid arthritis (Miyamae *et al*, 2006; Wilson *et al*, 2010). However, in the MI model, there were no differences in the levels of the inflammatory cytokines TNF-α and IL-1β in the ischemic hearts of cfKO compared to WT mice (Fig EV2A). Moreover, there was no significant difference of macrophage infiltration into infarcted lesion between WT and cfKO mice heart (Fig EV2B).

To further characterize the hearts of both strains of mice, apoptosis, microvasculature density, cardiac hypertrophy, and key intracellular signaling molecules were analyzed. TUNEL staining showed that the frequency of apoptotic cell death in vimentin-positive cardiac fibroblasts was not different between the WT and Fstl1-cfKO hearts after MI (Appendix Fig S5). Capillary density in the cardiac border zone was attenuated in cfKO than WT hearts in the post-MI condition (Appendix Fig S6), consistent with report of a pro-angiogenic effect of Fstl1 in hind limb ischemia model (Ouchi *et al*, 2008). In contrast, there was no significant difference of cross-sectional area of cardiomyocyte in the remote area between WT and cfKO mice heart (Appendix Fig S7). The deficiency of Fstl1 also led to reductions in the induction of both Akt Ser473 phosphorylation and AMPK Thr172 phosphorylation in the post-MI heart (Appendix Fig S8), consistent with previously published data showing that Fstl1 is upstream of these both of these signaling pathways in other contexts (Oshima *et al*, 2008; Ogura *et al*, 2012).

### TGF-β1 promotes Fstl1 expression in cardiac fibroblasts

It has been reported that TGF-β1 promotes Fstl1 expression in several cell types and organs (Shibanuma *et al*, 1993; Sundaram *et al*, 2013; Rainer *et al*, 2014). However, the regulation of Fstl1 by TGF-β1 cardiac fibroblasts and myofibroblasts has not been examined. Neonatal rat cardiac fibroblast (NRCFb) cells were treated with recombinant TGF-β1 protein (10 ng/ml), to promote differentiation to myofibroblasts, or vehicle, and Fstl1 expression was assessed by immunofluorescence and Western blotting (Fig 6). Cultured fibroblasts expressed Fstl1 at baseline, and Fstl1 levels increased upon TGF-β1-induced myofibroblast differentiation that was reflected by an increase in α-SMA expression (Fig 6A). The time course of Fstl1 induction by TGF-β1 is shown by immunoblot in Fig 6B. In addition to upregulating Fstl1, TGF-β1 also increased the transcript levels of S100a4 and α-SMA (Fig 6C).

### Fstl1 does not activate Smad signaling in the heart or cardiac fibroblasts

It has been previously reported that Fstl1 potentiates TGF-β1-induced Smad signaling in lung (Dong *et al*, 2015). Permanent LAD ligation led to increases in Smad2 and Smad3 phosphorylation at residues Ser465/467 and Ser423/425, respectively (Fig EV3A). Fstl1 deficiency led to a trend toward reduced Smad2 and Smad3 phosphorylation, but these differences were not statistically significant ($P = 0.32$ and $P = 0.056$, respectively, Tukey's test, $n = 4$ for each group). To analyze Fstl1-induced Smad phosphorylation in greater detail, NRCFbs were treated with recombinant Fstl1 in the presence or absence of recombinant TGF-β1. Whereas TGF-β1 led to marked

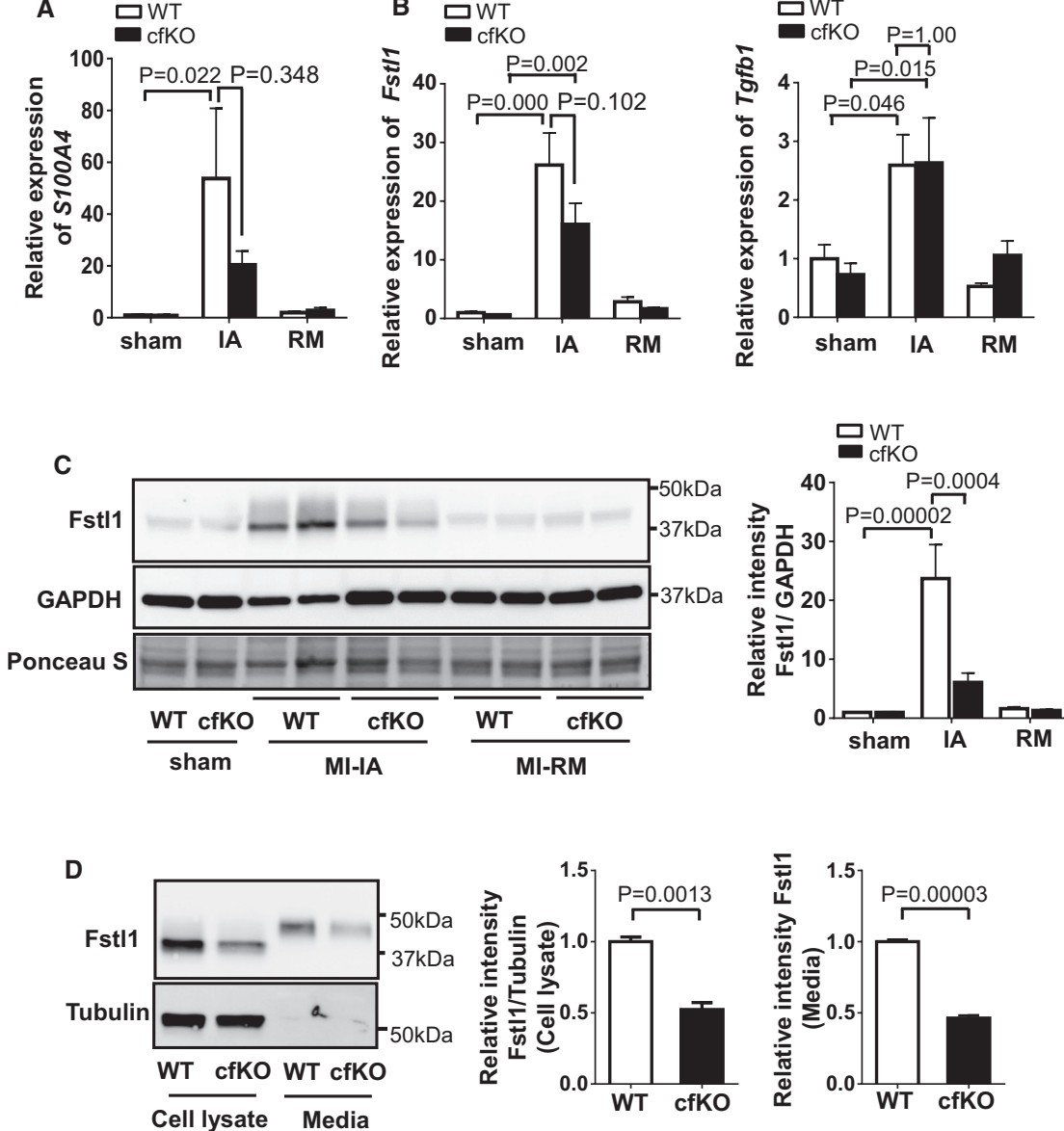

**Figure 3.  Fstl1 deletion in S100a4^cre+ × Fstl1^flox/flox mice.**
Genotypes are indicated as WT for S100a4^cre−/− × Fstl1^flox/flox mice and cfKO for S100a4^cre+/− × Fstl1^flox/flox.

A  S100a4 expression is induced in the ischemic area (IA). qPCR analysis of mRNA expression of *S100a4* in ischemic heart and sham-operated heart of WT mice and cfKO mice. Heart samples were harvested at 7 days after the surgery. Statistical analysis was performed by two-way ANOVA. *Post hoc* test was performed by Tukey's test. Error bars represent mean ± SEM (*n* = 16 and 15 for WT and cfKO sham group, *n* = 15 and 14 for WT and cfKO MI group, respectively).

B  qPCR analysis of mRNA expression of *Fstl1* and *Tgfβ1* in sham and post-MI heart. Error bars represent mean ± SEM (*n* = 16 and 15 for WT and cfKO sham group, *n* = 15 and 14 for WT and cfKO MI group, respectively). Statistical analysis was performed by two-way ANOVA. *Post hoc* test was performed by Tukey's test.

C  Western blot analysis of Fstl1 protein expression in ischemic and sham-operated hearts at day 7 after the surgery. Quantified values of Fstl1 protein in WT and cfKO mouse hearts normalized by GAPDH band intensity are shown. Statistical analysis was performed by two-way ANOVA. *Post hoc* test was performed by Tukey's test. Error bars represent mean ± SEM (*n* = 5 for each sham group and *n* = 6 for each MI group).

D  Fstl1 protein expression in isolated cardiac fibroblasts from cfKO and littermate WT neonatal mice. Cell lysate and its media from cells cultured for 24 h without FBS were assessed by Western blotting. Error bars represent mean ± SEM (*n* = 3 per group). Statistical analysis was performed by unpaired *t*-test (two-tailed). The experiments were performed twice independently.

Source data are available online for this figure.

increases in Smad2 and Smad3 phosphorylation, Fstl1 had no detectable effect on Smad phosphorylation in the presence or absence of TGF-β1 stimulation (Fig EV3B).

The treatment of NRCFbs with siRNA directed against Fstl1 led to the highly effective ablation of the protein under baseline and TGF-β1-stimulated conditions (Fig EV4A). At the level of mRNA,

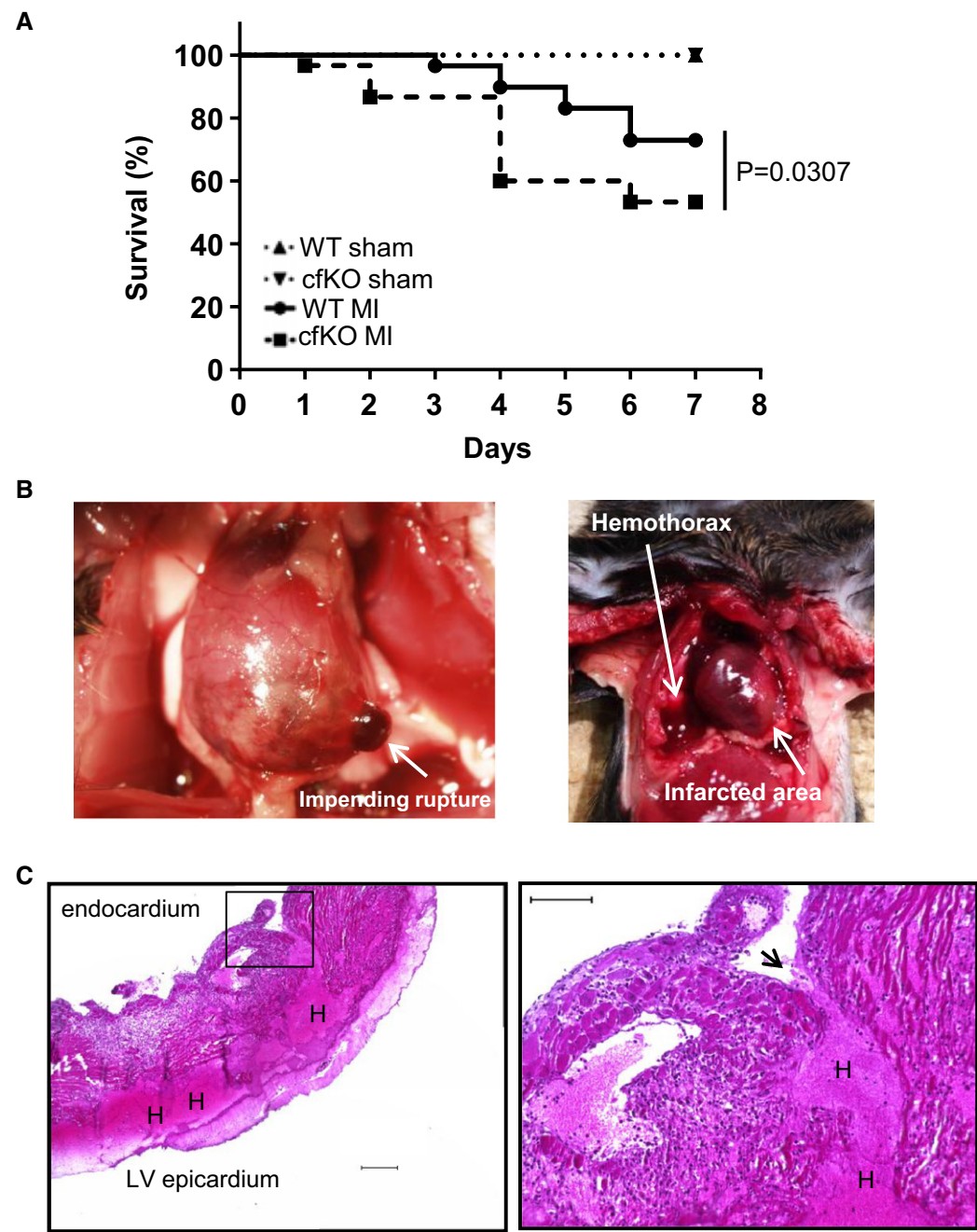

**Figure 4.  Higher mortality due to post-infarction cardiac rupture in S100a4$^{cre+/-}$ × Fstl1$^{flox/flox}$ mice.**

A  Mouse survival curve after MI and sham surgery. The mortality of the WT and Fstl1-cfKO mice after MI surgery was 27.1 and 46.7%, respectively. Log-rank (Mantel-Cox) test was used for statistical analysis ($n$ = 16 for WT sham, $n$ = 15 for Fstl1-cfKO sham, $n$ = 59 for WT MI and $n$ = 30 for Fstl1-cfKO MI).

B  Representative macroscopic images of cardiac rupture. Left: impending LV rupture found in a cfKO mouse post-MI when sampling. Right: evidence of hemothorax that was diagnosed as cardiac rupture in deceased mice.

C  Representative microscopic images of ruptured heart stained by H&E. Penetrating hematoma due to a tear from the LV endocardium to the epicardium. Left: an image obtained with a low magnification lens. Scale bar indicates 200 μm. Right: higher magnification image. Scale bar indicates 100 μm. Arrow indicates the tear site. H indicates hematoma.

Source data are available online for this figure.

the Fstl1 transcript was reduced by more than 99% (Fig EV4B). Ablation of endogenous Fstl1 by siRNA treatment had no detectable effects on Smad2 or Smad3 phosphorylation in either the presence or absence of TGF-β1 stimulation (Fig EV4C). We also assessed the consequences of Fstl1 ablation or supplementation on the expression of fibronectin, collagen I, and α-SMA, markers of myofibroblast

**Table 1.  Mortality after MI surgery.**

|  | Sham | | MI | |
| --- | --- | --- | --- | --- |
|  | WT | cfKO | WT | cfKO |
| Total mice number | 16 | 15 | 59 | 30 |
| Number of deaths (day 1 to day 7) | 0 | 0 | 16 | 14 |
| Cardiac rupture | 0 | 0 | 16 | 14 |
| Unknown cause of death | 0 | 0 | 0 | 1 |
| % of mortality until day 7 | 0 | 0 | 27.1 | 46.7 |
| Log-rank (Mantel–Cox) test |  | | $P = 0.0307$ | |
| Gehan–Breslow–Wilcoxon test |  | | $P = 0.0155$ | |
| Hazard ratio (95% CI of the ratio) |  | | 2.482 (cfKO/WT) (1.088 to 5.658) | |

differentiation that are regulated by Smad signaling (Fig EV4D). Manipulation of Fstl1 had no detectable impact on the levels of these marker proteins. Thus, studies with cultured cardiac fibroblasts did not reveal a direct modulatory activity of exogenous or endogenous Fstl1 on Smad signaling.

### Fstl1 activates fibroblast migration and proliferation via ERK1/2 signaling

The migration and proliferation of cardiac fibroblasts are critical in the infarct healing process, and extracellular signaling regulated kinase 1/2 (ERK1/2) plays an essential role in these processes (Takahashi et al, 2012; Clement et al, 2013; Sepe et al, 2013; Shaw et al, 2015). Recombinant Fstl1 led to the rapid and transient phosphorylation of ERK1/2 in serum-deprived NRCFbs (Fig 7A). The activating phosphorylation of ERK1/2 by Fstl1 was abolished by pretreatment with ERK inhibitor PD98059 (Fig 7B). Fstl1 ablation led to a significant reduction in NRCFb migration in a migration assay (Fig 7C), whereas exogenous Fstl1 protein stimulated migratory activity of Fstl1-knockdown fibroblasts (Fig 7D). Consistently,

treatment with PD98059 reversed the stimulatory effect of recombinant Fstl1 on migration.

Mesenchymal cell movement relies on focal substrate adhesion involving lamellipodia formation on the leading edge of the cell (Sixt, 2012). To assess the effect of Fstl1 on lamellipodia, NRCFbs were stained with an immunofluorescent, phalloidin-conjugated antibody against F-actin. Treatment with recombinant Fstl1 markedly changed fibroblast cell morphology (Fig 7E). Changes in the cytoskeleton were notable and lamellipodia formation was apparent at 3 h following Fstl1 addition. These Fstl1-induced alterations were reversed by treatment with PD98059.

Knockdown of Fstl1 by siRNA led to a reduction in NRCFb proliferation as assessed by EdU incorporation (Fig 8A). Conversely, treatment with exogenous Fstl1 protein increased cardiac fibroblast proliferation (Fig 8B). In separate experiments, treatment with PD98059 was found to block the stimulatory effect of exogenous Fstl1 on fibroblast proliferation (Fig 8C).

Fstl1 is a glycosylated protein and the degree of this post-translational modification is dependent upon it cellular source and its secretion (Oshima et al, 2008; Wei et al, 2015). Comparing Fstl1 produced by cultured cardiac fibroblasts and cardiac myocytes in the cell lysate and media revealed immunogenic peptides of varying electrophoretic mobilities (Fig EV5A). Treatment with tunicamycin converted these forms to a fast electrophoretic mobility form suggesting that the heterogeneity resulted from differences in Fstl1 glycosylation. Recently, it was reported that different isoforms of glycosylated Fstl1 protein produced from cardiomyocyte and epicardial mesothelial cells are functionally different with regard to their ability to stimulate cardiac myocyte proliferation (Wei et al, 2015). To test whether the different glycosylation isoforms of Fstl1 differentially effect fibroblast activation, we compared Fstl1 produced from mammalian cells, insect cells, and E. coli cells (Fig EV5B and Table EV5). Although these Fstl1 isoforms differ widely in their degree of glycosylation, no differences were observed in their ability to promote the migration of cardiac fibroblasts (Fig EV5C).

**Table 2.  Functional analysis of Fstl1-cfKO and WT mice after MI. Cardiac function was assessed by echocardiography at 6 days after the surgery. Mouse blood pressure was measured at 5 days after the surgery by the tail cuff method.**

|  | Sham | | MI | |
| --- | --- | --- | --- | --- |
|  | WT | cfKO | WT | cfKO |
|  | (n = 16) | (n = 14) | (n = 15) | (n = 14) |
| FS (%) | 56.83 ± 2.64 | 47.49 ± 1.84 | 21.47 ± 3.51[a] | 23.07 ± 3.41[a] |
| LVIDd (mm) | 3.26 ± 0.11 | 3.39 ± 0.07 | 4.59 ± 0.17[a] | 4.50 ± 0.19[a] |
| LVIDs (mm) | 1.45 ± 0.12 | 1.78 ± 0.08 | 3.69 ± 0.26[a] | 3.52 ± 0.27[a] |
| IVSd (mm) | 0.91 ± 0.05 | 0.86 ± 0.04 | 0.55 ± 0.05[a] | 0.49 ± 0.03[a] |
| PWd (mm) | 0.90 ± 0.04 | 0.82 ± 0.02 | 0.95 ± 0.12 | 0.78 ± 0.05 |
| HW/TL (mg/mm) | 7.42 ± 0.52 | 6.59 ± 0.21 | 8.07 ± 0.46 | 8.04 ± 0.39[a] |
| LW/TL (mg/mm) | 7.69 ± 0.49 | 7.17 ± 0.21 | 10.6 ± 1.14[a] | 8.69 ± 0.99 |
| Systolic BP (mmHg), day 5 | 129.4 ± 3.62 | 127.6 ± 4.92 | 113.6 ± 6.35 | 122.4 ± 5.9 |
|  | (n = 6) | (n = 6) | (n = 6) | (n = 5) |

FS, fractional shortening; LVIDd, left ventricular internal diastolic dimension; LVIDS, left ventricular internal systolic dimension; IVSd, intraventricular septum diastole; HW, heart weight; TL, tibia length; and LW, lung weight.
The statistical analysis was performed by two-way ANOVA. Post hoc analysis was performed by Tukey's test. Data are shown as mean ± SEM.
[a]$P < 0.05$ in MI versus sham.

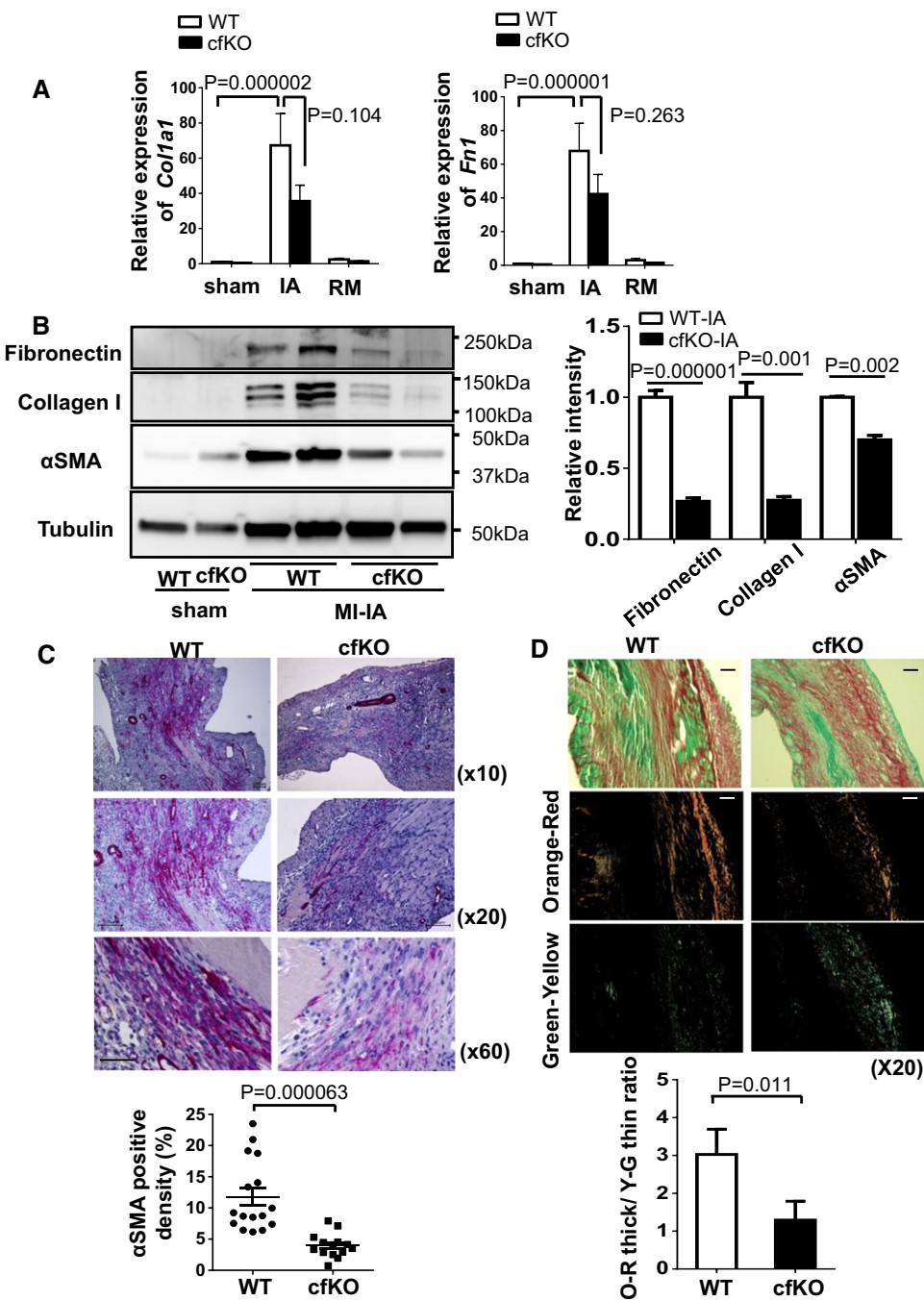

**Figure 5. Impaired ECM protein synthesis in infarcted area of S100a4^cre+/− × Fstl1^flox/flox mice.**

A Transcript mRNA expression of *Col1a1* and *Fn1* in sham and post-MI hearts. Hearts were obtained at day 7 after surgery. Error bars represent mean ± SEM
(*n* = 13–16 for each group). Statistical analysis was performed by two-way ANOVA. *Post hoc* test was performed by Tukey's test.

B Protein expression of collagen I, fibronectin and α-SMA in the infarcted and sham hearts at day 7 after surgery, as assessed by Western blotting. Quantified values of
proteins of interest were normalized relative to the tubulin band intensity. Error bars represent mean ± SEM. Statistical analysis was performed by unpaired *t*-test
(two-tailed) for fibronectin and α-SMA and nonparametric *t*-test (two-tailed) for collagen I (fibronectin and collagen: *n* = 5 per group, α-SMA: *n* = 3 and 4 for WT-IA
and cfKO-IA, respectively).

C α-smooth muscle actin-positive myofibroblasts (red) were stained in histological sections from infarcted hearts of WT and cfKO mice. Samples were harvested at day
7 after MI. Counter staining was performed by hematoxylin. Scale bar indicates 100 μm for top and middle images and 50 μm for bottom images. Error bars
represent mean ± SEM. Statistical analysis was performed by Mann–Whitney *U*-test (*n* = 16 for WT and *n* = 14 for cfKO).

D Representative images of Picrosirius red staining of infarcted heart at 7 days after MI surgery. Images were taken by standard light (top) and polarized light
microscopes. Scale bar indicates 100 μm. Polarized collagen fibers were sorted to orange-red (middle) and green-yellow (bottom) fibers by ImageJ software. Error bars
represent mean ± SEM. Statistical analysis was performed by unpaired *t*-test (two-tailed) (*n* = 11 for WT and *n* = 13 for cfKO).

Source data are available online for this figure.

## Discussion

Although the community incidence of ST elevation myocardial infarction (STEMI) has been decreasing, complications of acute MI remain as major life-threatening problems that require intensive treatment (Yeh *et al*, 2010). Left ventricular free wall rupture is reported to occur in 1–6% of MI patients (Pappas *et al*, 1991; Becker *et al*, 1996; Slater *et al*, 2000; Figueras *et al*, 2008), and mortality is reported to be as high as 60% in patients with this condition (O'Gara *et al*, 2013). Here, we show that cardiac fibroblasts are a functionally significant source of the secreted glycoprotein Fstl1 in a murine model of MI and that the loss of Fstl1 from this cellular compartment leads to an increase in mortality due to cardiac rupture in the acute phase. This phenotype was characterized by a diminished number of myofibroblasts and reductions in the extracellular matrix proteins collagen I and fibronectin in the infarct area. Reduced Fstl1 expression by fibroblasts also led to a more loosely packed collagen arrays. Collectively, these data reveal the importance of cardiac fibroblast-derived Fstl1 in the synthesis and maturation of extracellular matrix, contributing to infarct remodeling in response to acute MI.

Mechanistically, we found that Fstl1 triggers ERK1/2 signaling in cardiac fibroblasts, and this signaling step is essential for Fstl1-mediated activation of fibroblast proliferation and migration. In contrast, in cultured cardiac fibroblasts, Fstl1 had no detectable impact on Smad2/3 activation or on the expression of α-smooth muscle actin, fibronectin, or collagen I, or on the abilities of TGF-β1 to activate these pathways. Thus, we speculate that the reparative fibrotic response conferred by Fstl1 *in vivo* is the consequence of its early activation and mobilization of cardiac fibroblasts that leads to greater myofibroblast accumulation in the infarct area and, as a consequence, greater synthesis and maturation of the extracellular matrix (see model in Fig 8D). In other models of cardiac pathology, Fstl1 has been shown to have anti-hypertrophic and anti-inflammatory activities (Shimano *et al*, 2011; Ogura *et al*, 2012). However, Fstl1 ablation in fibroblasts did not detectably affect these parameters in the current study, perhaps because the prohypertrophic and pro-inflammatory signals in the post-MI heart overwhelm any impact of partial Fstl1 deficiency.

Fstl1 is categorized as a member of the follistatin family of proteins because it contains a follistatin-like domain. Other members of this family, that is follistatin and Fstl3, function as extracellular antagonists of TGF-β superfamily proteins. However, Fstl1 has relatively low sequence homology with follistatin and Fstl3. Fstl1 has considerable homology with SPARC (identity 25.0%, similarity 51.6%) that functions as an early regulator of extracellular matrix maturation following MI (McCurdy *et al*, 2011) and whose genetic deficiency leads to post-MI cardiac rupture in mice (Schellings *et al*, 2009). Rather than serving as an extracellular agonist of TGF-β1 signaling, it has been shown both here and previously (Oshima *et al*, 2008; Ouchi *et al*, 2008) that recombinant Fstl1 can rapidly activate intracellular signaling pathways in cells that are deprived of mitogens and that its direct action as a ligand can be mediated by a cell-surface receptor (Ouchi *et al*, 2010). In contrast to the results reported here, Dong *et al* have reported that inhibition of Fstl1 attenuates bleomycin-induced pulmonary fibrosis (Dong *et al*, 2015). Their study showed that Fstl1 activates fibroblasts by potentiating TGF-β1-stimulated Smad2/3 signaling and inducing fibrosis. In contrast, our study could find no direct involvement of Fstl1 in Smad2/3 activation, suggesting that Fstl1's actions on fibroblasts can be context dependent. In this regard, it may be relevant that the pulmonary fibrosis model examined the systemic haplodepletion of Fstl1, whereas this study examined Fstl1 depletion restricted to a portion of cardiac fibroblasts.

Prior studies have documented the cardiovascular-protective properties of Fstl1 supplementation in experimental MI (Ogura *et al*, 2012; Wei *et al*, 2015). The current study reports, for the first time, the essential role of endogenous Fstl1 in the ischemic myocardium using genetic loss-of-function methodology and elucidates a novel role for Fstl1 in cardiac fibroblast biology. In the fetus, Fstl1 expressed in mesenchymal cells plays a crucial role for organ development (Sylva *et al*, 2011, 2013; Geng *et al*, 2013). In the adult, Fstl1 tends to be expressed in various mesenchymal lineage cells including cardiomyocytes (Shimano *et al*, 2011), skeletal muscle cells (Ouchi *et al*, 2008; Gorgens *et al*, 2013; Miyabe *et al*, 2014), osteocytes (Wilson *et al*, 2010), chondrocytes (Wilson *et al*, 2010), fibroblasts (Wilson *et al*, 2010), smooth muscle cells (Liu *et al*, 2006), and pre-adipocytes (Wu *et al*, 2010). In this study, we have identified fibroblasts and myofibroblasts, that is non-cardiomyocyte interstitial cells, as the main source of the large induction of Fstl1 after MI and demonstrated that this source of Fstl1 is essential for a reparative fibrotic response. On the other hand, cardiac myocyte-derived Fstl1 has been shown to be functionally significant in antagonizing myocyte hypertrophic growth in a model of left ventricular hypertrophy (Shimano *et al*, 2011). It has also been shown that epicardial cells represent a functionally significant source of Fstl1 that is lost following MI due to its downregulation from this compartment (Wei *et al*, 2015). Conversely, the delivery of a hypoglycosylated form of Fstl1 from an epicardial matrix patch is reported to promote the regeneration of cardiac myocytes in the adult mammalian heart post-MI. While Fstl1 has been shown to exert a protective role in every cardiovascular injury model analyzed to date, the divergent phenotypes observed in these models may reflect differences in its spatial or temporal expression patterns. It is also possible that quantitative differences in Fstl1 induction levels in the different cardiac injury models have divergent effects on various cardiac cell types that might differ in their threshold responses to this ligand.

With regard to the cardiac rupture phenotype, we find that Fstl1 levels are appreciably reduced in the post-MI heart of S100a4$^{cre+/-}$ × Fstl1$^{flox/flox}$ mice, but these mice display no reduction in Fstl1 levels at baseline where fibroblasts are inactive and have a minor contribution to the total mass of the heart. In contrast, mouse genetic ablation experiments show that neither cardiac myocytes nor macrophages significantly contribute to the very high levels of Fstl1 that are detected in the post-MI heart (approximately an 80-fold upregulation of Fstl1 protein expression by 7 days). Thus, we conclude that the cardiac rupture phenotype of the S100a4$^{cre+/-}$ × Fstl1$^{flox/flox}$ strain results from significantly diminished Fstl1 expression leading to diminished paracrine or autocrine protective actions. Alternatively, the Fstl1 produced by distinct cellular sources could be subject to differences in post-translational glycosylation modifications that may yield functionally different isoforms as proposed by others (Hambrock *et al*, 2004; Wei *et al*, 2015). However, we did not find differences between the abilities of recombinant Fstl1 protein produced by mammalian, insect, or bacterial cells to promote the activation of cardiac fibroblast *in vitro*, although these sources of protein differed widely in their degrees of

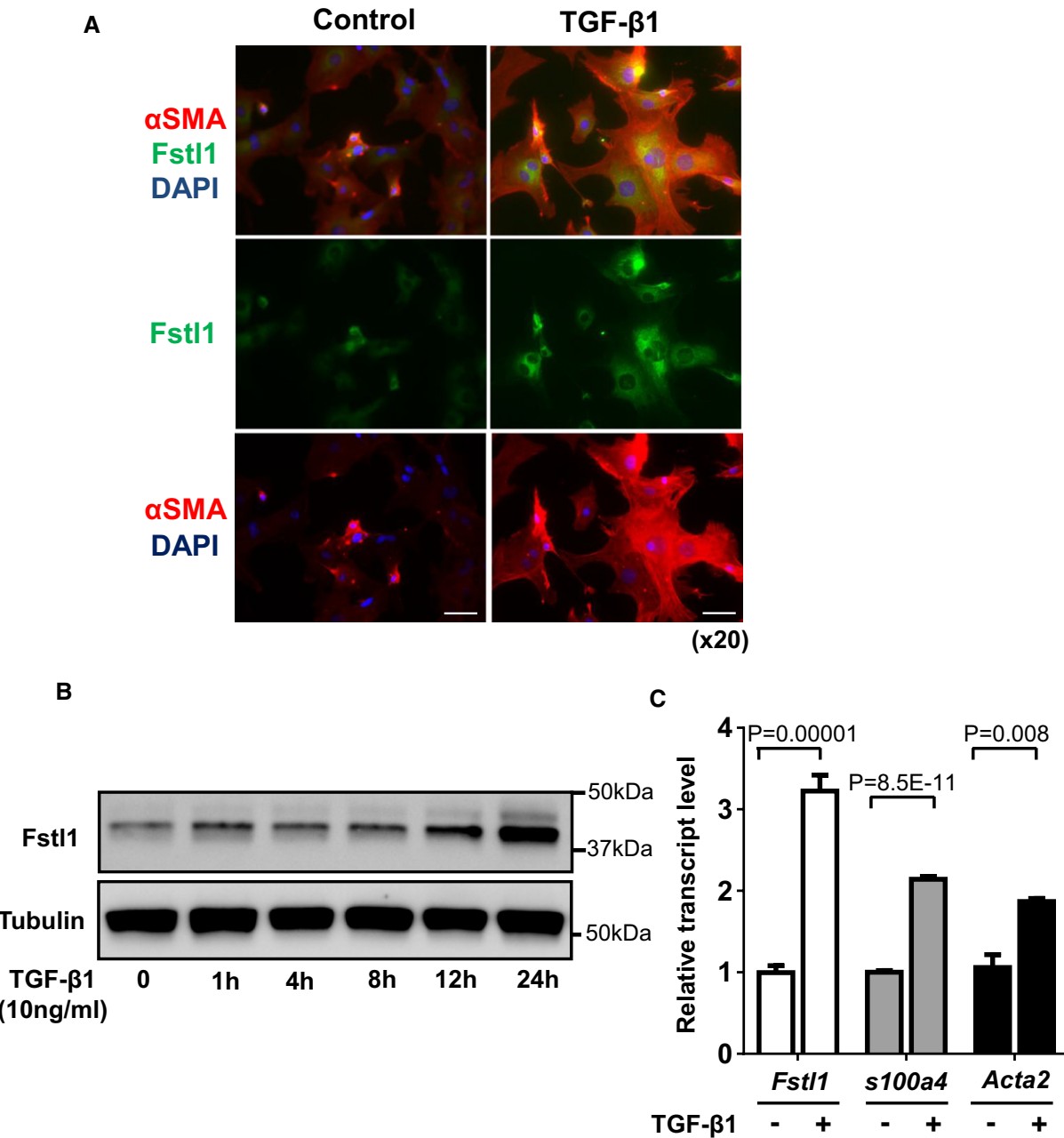

**Figure 6. TGF-β1 promotes Fstl1 induction in cardiac fibroblasts.**

A Immunofluorescence staining of Fstl1 (green) and α-SMA (red) in NRCFbs at 24 h after recombinant TGF-β1 (10 ng/ml) stimulation. Nuclei were stained by DAPI. Scale bar indicates 50 μm.

B Time course changes of Fstl1 protein expression in NRCFbs after stimulation of recombinant TGF-β1 (10 ng/ml). Protein expression of Fstl1 was detected by immunoblotting. Tubulin was used as a loading control (*n* = 3 for each time point). Two independent experiments were performed.

C Transcript level of *Fstl1, S100a4,* and *Acta2* mRNA in NRCFbs was determined by qPCR analysis. The samples were harvested at 24 h after stimulation with recombinant TGF-β1 (10 ng/ml) or control vehicle. Error bars represent mean ± SEM (*Fstl1* and *S100a4: n* = 6 for each group, *Acta2: n* = 3 for each group). Statistical analysis was performed by unpaired *t*-test (two-tailed) for *Fstl1* and *S100a4,* and nonparametric unpaired *t*-test (two-tailed) for Acta2. Two independent experiments were performed.

Source data are available online for this figure.

glycosylation. Regardless, additional investigations to dissect the tissue-specific, spatiotemporal, and structure–function relationships of Fstl1 are warranted.

A limitation of this study is that only a partial deficiency of cardiac fibroblast Fstl1 could be achieved using the S100a4$^{\text{cre}+/-}$ murine strain. Fstl1 is strongly induced in the heart following LAD

**Figure 7. Fstl1 promotes the phosphorylation of ERK1/2 and cardiac fibroblast migration.**

A   NRCFbs at passage 1 were stimulated with recombinant Fstl1 (50 ng/ml) or vehicle after cultured in serum-reduced conditions (FBS 0.5%) for 24 h. The samples were harvested at the indicated time points after stimulation. The expression of ERK1/2 and tubulin were detected by immunoblotting. Error bars represent mean ± SEM. Statistical analysis was performed by one-way ANOVA and Tukey's multiple comparison test ($n = 3$ for each time point). Three independent experiments were performed.

B   Fstl1 induced phosphorylation of ERK1/2 was ablated by pretreatment with ERK inhibitor PD98059. NRCFbs were treated by PD98059 (5 nM) for 30 min and then stimulated with recombinant Fstl1 protein (50 ng/ml) for 15 min. Error bars represent mean ± SEM. Statistical analysis was performed by one-way ANOVA and Tukey's multiple comparison test ($n = 3$ for each time point). Two independent experiments were performed.

C   Representative images of NRCFbs scratch assay (left) and quantified cell migration (right). NRCFbs were transfected by Fstl1 siRNA or siRNA non-targeting negative control for 12 h followed by culturing in 0.5% FBS media for 24 h. The confluent cell sheet was scratched and cell migration was assessed at 6 h after the scratch. Scale bar indicates 100 μm. Error bars represent mean ± SEM ($n = 10$ for each group). Statistical analysis was performed by unpaired $t$-test (two-tailed). Two independent experiments were performed.

D   Fstl1 stimulation of cell migration was reversed by PD98059. Endogenous Fstl1 in NRCFbs was ablated by Fstl1 siRNA. Serum-deprived NRCFbs were treated with PD98059 (5 nM) for 30 min and then stimulated by Fstl1 (50 ng/ml) or vehicle. Cell migration was assessed at 6 h after Fstl1 stimulation. Error bars represent mean ± SEM ($n = 7$–9 for each group). Statistical analysis was performed by Kruskal–Wallis test and Dunnett's T3 test. Two independent experiments were performed.

E   Morphological changes of NRCFbs after recombinant Fstl1 stimulation was assessed by immunocytochemistry. Twenty-four-hour serum-starved NRCFbs were stimulated by Fstl1 (50 ng/ml) or vehicle for 3 h. PD98059 (5 nM) or DMSO was added 30 min prior to Fstl1 stimulation. Cells were stained with Alexa Fluor 488-conjugated phalloidin antibody, and nuclei were stained by DAPI. Arrow shows lamellipodium of the cell. Scale bar indicates 50 μm.

Source data are available online for this figure.

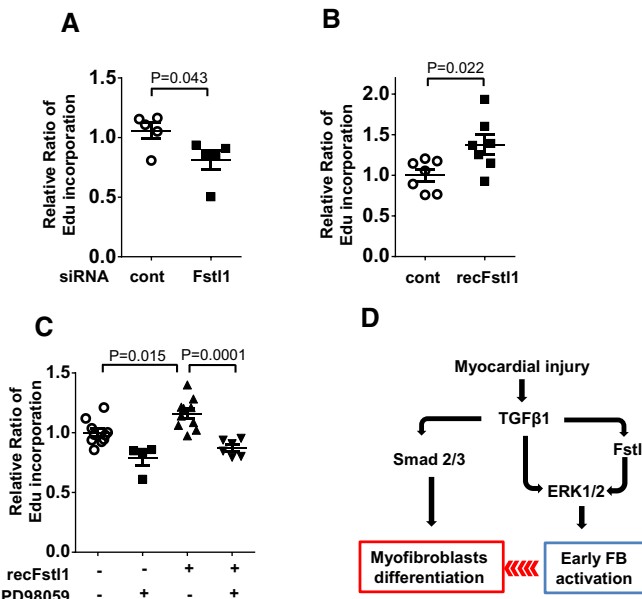

**Figure 8. Fstl1 promotes cardiac fibroblasts proliferation via the ERK1/2-dependent pathway.**

A, B  Fibroblast proliferation was assessed by Edu incorporation assay. Endogenous Fstl1 was ablated by siRNA. NRCFbs were cultured in 0.5% FBS condition for 48 h to synchronize the cell cycle. EdU (10 μM as the final concentration) was added into media at 4 h before harvest. Error bars represent mean ± SEM (n = 5, each group). Statistical analysis was performed by unpaired t-test (two-tailed). Two independent experiments were performed. (B) Effect of exogenous Fstl1 on cardiac fibroblast proliferation. Cells were cultured in FBS 0.5% media for 24 h. Recombinant Fstl1 (50 ng/ml) or vehicle was added to 2% FBS-containing media and cultured for 48 h. EdU was added into media at 4 h before harvest. Error bars represent mean ± SEM (n = 7, each group). Statistical analysis was performed by unpaired t-test (two-tailed). Two independent experiments were performed.

C  Fstl1-promoted fibroblast proliferation was diminished by PD98059. PD98059 (5 nM) was added 30 min prior to recombinant Fstl1 stimulation (50 ng/ml). The cells were cultured for 48 h, and EdU was added into media at 4 h before harvest. Error bars represent mean ± SEM (n = 4–10, per treatment group). Statistical analysis was performed by one-way ANOVA and Tukey's multiple comparison test. Two independent experiments were performed.

D  Schema of the role of Fstl1 in infarct repair. Fstl1 is upregulated by TGF-β1 in infarcted heart, and it contributes to the proliferation and migration of fibroblasts into the infarcted site and border zone. These actions increase the number of myofibroblasts (MyoFB) in the infarcted lesion. In turn, fibroblasts and myofibroblasts synthesize ECM components including collagens and fibronectin and protect the infarcted heart from rupture.

Source data are available online for this figure.

Fstl1- and S100A4-positive cells. Similarly, we found partial colocalization of Fstl1 with other fibroblast marker proteins including vimentin. Cardiac myofibroblasts are thought to arise from resident fibroblasts, endothelial cells, smooth muscle cells, and hematopoietic progenitor cells (Haudek et al, 2006; Humphreys et al, 2010; Zeisberg & Kalluri, 2010; Shinde & Frangogiannis, 2014). Thus, there is a high degree of heterogeneity among cardiac fibroblasts (Chen & Frangogiannis, 2013; Moore-Morris et al, 2014), and the quantitative ablation of genes in this cell type represents a vexing problem for the field. Regardless, the partial ablation of Fstl1 from the cardiac fibroblast pool led to notable differences in myofibroblast abundance, matrix synthesis and maturation, and frequency of cardiac rupture post-MI, providing evidence for the importance of this protein in the healing response to acute MI.

Clinical studies have documented the clinical relevance of Fstl1 in heart failure and ACS (Lara-Pezzi et al, 2008; El-Armouche et al, 2011). Of interest, Fstl1 has been shown to function in a "biomarker network" with growth differentiation factor 15 (GDF15) in the pathophysiology of ACS (Widera et al, 2012). Experimental studies showed that Fstl1 is both essential and sufficient for a full GDF15 protective effect in in vitro and in vivo models. Of relevance to this study, systemic GDF15 deficiency leads to increased cardiac rupture in the murine MI model (Kempf et al, 2011). In contrast to Fstl1, GDF15 protects against cardiac rupture via an anti-inflammatory mechanism involving the suppression of polymorphonuclear leukocyte recruitment. Collectively, these data underscore the importance of this biological pathway in the early MI healing process, given the evidence that GDF15 and Fstl1 function as clinically relevant orthogonal biomarkers.

Left ventricular free wall rupture is a serious complication of the acute phase following MI (O'Gara et al, 2013). Currently, the optimal treatment for these patients is highly invasive and consists of the surgical attachment of an overlay patch to the affected region with sutures (Nasir et al, 2014). Thus, a better understanding of the endogenous repair properties of the heart could lead to the development of pharmacological agents to avoid acute cardiac rupture or lead to less invasive methodologies to repair the defect. Here, we show that the induction of Fstl1 expression in cardiac fibroblasts is essential for a reparative fibrotic response following acute MI in mice. Partial Fstl1 deficiency in cardiac fibroblasts led to increased cardiac rupture, and surviving mice displayed a reduced extracellular matrix synthesis and maturation and a reduced abundance of myofibroblasts in the infract zone. Mechanistically, it is shown that Fstl1 plays a role in the early stages of fibroblast activation involving the proliferation and migration of these cells. Because cardiac fibroblasts are the major source of Fstl1 production in the ischemic myocardium, it is reasonable to speculate that Fstl1 participates in a positive feedback loop that is essential for the acute repair of the infarcted heart and that Fstl1 may have utility for the treatment or prevention of cardiac rupture.

# Materials and Methods

### Generation of conditional Fstl1-deficient mice and MI model

Mice of Fstl1 rendered in fibroblasts lineage were generated by crossing S100a4$^{cre+/-}$ transgenic mice and Fstl1$^{flox/flox}$ mice. S100a4$^{cre+/-}$ transgenic mice in BALB/c background were purchased

ligation, as is S100a4. In the S100a4$^{cre+/-}$Fstl1$^{flox/flox}$ strain, the level of Fstl1 induction was reduced by 40 and 70% at the levels of transcript and protein, respectively, within the infarcted tissue. A potential reason for this partial reduction is the concomitant expression of Fstl1 by other cellular components of the heart, as discussed above. However, Fstl1 produced by cardiac fibroblasts isolated from cfKO hearts was also reduced by ~50%, indicating a partial ablation of this gene from this cellular source. Accordingly, histological analysis of infarct zone fibroblasts revealed a partial colocalization of

                                    

from Jackson Laboratory (Bar Harbor, ME, USA). Fstl1$^{flox/flox}$ mice in FVB background (Sylva *et al*, 2011) were backcrossed with C57BL/6 mice in the animal facility in Boston University Medical Campus. S100a4$^{cre+/-}$ × Fstl1$^{flox/flox}$ mice were used as fibroblasts lineage Fstl1-deleted mice (Fstl1-cfKO), and their littermates S100a4$^{cre-/-}$ × Fstl1$^{flox/flox}$ mice were used as control wild type mice (WT mice). All male mice used in this study were 9–12 weeks old. Experimental rodent LAD ligation surgery and sham surgery were performed as previously described (Oshima *et al*, 2008). Briefly, following anesthetization (isoflurane inhalation) and tracheal intubation, the left coronary artery was ligated tightly with an 8-0 silk suture. Myocardial ischemia was confirmed by ST-T segment changes in ECG and color change of coronary flow occulted segment of left ventricle. All study protocols were approved by the Institutional Animal Care and Use Committee (IACUC) of Boston University. Operated mice were monitored every day, and cause of death was examined by internal examination. Death within 12 h after surgery was excluded from analysis. Cardiac rupture was defined by evidences of hemothorax or hematoma in left ventricular outer wall.

### Echocardiographic and blood pressure measurement

Cardiac function was assessed at 6 days after the surgery by Vevo770 machine using RMV 707B probe (VisualSonics, Fujifilm). Left ventricular internal diastolic dimension (LVIDd) and left ventricular internal systolic dimension (LVIDs) were measured from M-mode images obtained by left ventricular short-axis view. The averages of three to five measured values per mouse by echocardiography were used for statistical analysis. Mouse blood pressure was measured at 5 days after the surgery by tail cuff method using BP-2000 Blood Pressure Analysis System™ (Visitech Systems, Inc.). The median of systolic blood pressure was calculated from 15 to 20 measured values per each mouse.

### Isolation of cardiac fibroblasts and cell culture

Primary neonatal rat cardiac fibroblasts (NRCFbs) were isolated from 1- to 2-day-old Sprague Dawley rats (Charles River Laboratory). To validate the ablation of Fstl1 in cardiac fibroblasts in Fstl1-cfKO, 1- to 2-day-old neonatal mice with genotyping of S100a4$^{cre+/-}$ × Fstl1$^{flox/flox}$ and S100a4$^{cre-/-}$ × Fstl1$^{flox/flox}$ were used. Animals were euthanized under the approved protocol. Harvested hearts were washed by sterile PBS and minced in ice-cold PBS. Minced heart tissues were digested with digestion buffer prepared with 0.05% (w/v) collagenase type II (Worthington Biochemical) and 0.06% (w/v) pancreatin (Sigma-Aldrich) in PBS. Tissue digestion was performed by incubating minced hearts in digestion solution for 10 min at 37°C. Supernatant was collected into sterile falcon tubes via 70-μm filter, and tissue digestion was repeated for five times with freshly added digestion buffer. The collected cells were spun down and resuspended in culture medium: DMEM/F-12 with 10% HI-FBS (both Life Technologies) and penicillin–streptomycin (Thermo Scientific). After washing out digestion buffer with culture medium twice, cells were seeded on cell culture dishes and incubated for 1 h in the incubator (37°C, 5% $CO_2$, 90% humidity). After 1-h incubation, floating cells and culture medium were removed and attached cells were continued to culture with fresh culture medium. NRCFbs at passage 1–2 were used for this study.

To knock down endogenous Fstl1, On-TARGET plus rat SMART pool siRNA Fstl1 (Dharmacon, GE Healthcare) were transfected to NRCFbs by reverse transfection method using lipofectamine RNAimax (Life Technologies). On-TARGET plus non-targeting pool siRNA was used for control. Each *in vitro* assay was performed after culturing NRCFbs in FBS reduced media (0.5% FBS) for 24 h.

### Cell migration assay

NRCFbs at passage 1 were seeded in 12-well plates at $2 \times 10^5$ cells/well density. siRNA transfection was performed as described earlier. Following FBS starvation for 24 h, cell sheets were scratched by 20-μl pipette tip (Rainin). Cells were incubated with or without 50 ng/ml recombinant Fstl1 protein (PROTEOS, Inc., Kalamazoo, MI). ERK inhibitor PD98059 (5 nM, Sigma-Aldrich) or dimethyl sulfoxide (DMSO) as control was added to the medium 30 min prior to recombinant Fstl1 protein stimulation. Images were captured at 6 h after the scratching by Keyence BZ-9000 microscope (Keyence) using phase-contrast lens. Cell's migrated area was measured by BZ-II analyzer (Keyence) in 8–10 independent images per experimental group.

### Cell proliferation assay

NRCFbs at passage 1 were seeded in 96-well plates at $1 \times 10^4$ cells/well density. After overnight incubation (37°C, 5% $CO_2$, 90% humidity), siRNA transfection was performed by using lipofectamine RNAimax and Optimem medium. Eight hours later, medium was switched to DMEM/F-12 containing 0.5% FBS and cultured for additional 24 h to synchronize cell cycle stage. Recombinant Fstl1 protein (50 ng/ml) or control vehicle (PBS) was added to cell culture media (changed to FBS 2% contained DMEM/F-12) and cultured for 48 h. MEK inhibitor PD98059 (5 nM, Sigma-Aldrich) or DMSO as control was added to media 30 min prior to recombinant Fstl1 stimulation. EdU (Life Technologies) at 10 μM as the final concentration was added into media at 4 h before harvesting cells. Edu incorporation was detected by Click-iT Edu Microplate Assay by following manufacture's protocol (Life Technologies). Fluorescence intensity was measured by SpectraMax M3 (Molecular Devices).

### Production of recombinant Fstl1 protein

Custom-made recombinant human Fstl1 protein was produced in PROTEOS, Inc. A codon-optimized synthetic cDNA encoding human Fstl1 was produced and subcloned into pTT5 expression plasmid. The plasmid was transfected to HEK293 cells using linear PEI. Transfected HEK293 cells were cultured in shake flasks at 37°C in a humidified 5% $CO_2$ environment. Culture was harvested according to cell density and viability, and the conditioned supernatant was clarified by centrifugation. The protein purification was performed by batch Ni sepharose excel chromatography using Ni sepharose excel (GE Healthcare) followed by size exclusion chromatography (Superdex 200, GE Healthcare). Expression of Fstl1 protein was validated in cell lysate and purified protein using SDS–PAGE with Fstl1-specific antibody. The production of Fstl1 from Sf9 insect cells was described previously (Ouchi *et al*, 2010).

## Western blot analysis

The proteins from mouse hearts or cell lysates were homogenized in RIPA buffer (Sigma-Aldrich) containing phosphatase inhibitor (Phos-STOP®, Roche) and protease inhibitor (Complete Mini EDTA-free®, Roche). The protein concentration was measured using BCA Protein Assay Kit (Thermo Scientific). The absorbance was measured by SpectraMax M3 (Molecular Devices). Equal amounts of denatured protein samples (10–15 µg) were resolved by SDS–PAGE. Serum samples were denatured and loaded at 1 µl per well. The protein-transferred PVDF membranes (Bio-Rad) were incubated with following primary antibodies at 4°C overnight: anti-mouse Fstl1 antibody (polyclonal goat IgG, catalog #AF1738, R&D Systems, 1:1,000 dilution), α-SMA antibody (clone 1A4, catalog #A2547, Sigma-Aldrich, 1:5,000 dilution), collagen I antibody (catalog #ab21286, Abcam, 1:1,000 dilution), and fibronectin antibody (H-300, catalog #sc-9068, Santa Cruz Biotechnology, 1:1,000 dilution). Phospho-Smad2 (S465/467) (clone 138D4, catalog#3108, 1:1,000 dilution), phosho-Smad3 (S423/425) (clone C25A9, catalog#9520, 1:1,000 dilution), Smad 2 (clone 86F7, catalog#3122, 1:1,000 dilution), Smad 3 (clone C67H9, catalog#9523, 1:1,000 dilution), phospho-ERK1/2 (Thr202/Tyr204) (clone D13.14.4E, catalog#4370, 1:2,000 dilution), ERK1/2 (catalog#9102, 1:1,000 dilution), GAPDH (D16H11, catalog#8884, 1:1,000 dilution), and tubulin antibodies (catalog#8884, 1:1,000 dilution) were purchased from Cell Signaling Technology. The membrane was incubated with HRP-conjugated secondary antibodies of anti-mouse IgG, anti-rabbit IgG (Cell Signaling), or anti-goat IgG (Santa Cruz) for 1 h at room temperature. The specific proteins were detected by chemiluminescence substrate WesternBright™ Quantum (Advansta). Ponceau S staining (Sigma-Aldrich) was used to reveal equal protein amount loading. The images were obtained by ImageQuant LAS4000 (GE Healthcare), and band intensities were analyzed by using Image-Quant TL software (GE Healthcare).

## Histology, immunohistochemistry and immunofluorescent staining

The heart tissues were obtained at 7 days after LAD ligation or sham surgery. The samples were processed for paraffin embedding or OCT embedding depending on the purpose of staining. Picrosirius red staining, DAB substrate staining (Fstl1), and alkaline phosphatase substrate staining (α-SMA) were performed with paraffin-embedded sections. Heart tissues were fixed in 10% formalin overnight and embedded in paraffin. About 6-µm-thick sections were deparaffinized and rehydrated. For Picrosirius red staining, sections were incubated with freshly prepared staining buffer (1.2%/w picric acid in water, 0.1%/w Fast Green FCF and 0.1%/w Direct Red 80 solved in PBS) for 1 h at room temperature (all products from Sigma-Aldrich). Then, sections were washed briefly in dH2O and dehydrated. The slides were mounted by coverslip using Permount mounting medium (Fisher Scientific). The birefringence image of collagen was taken by Olympus BX40 microscope using U-POT drop in polarizer (Olympus Optical). The images were analyzed by ImageJ software (NIH) as previously described (Deguchi *et al*, 2005). Each color image of polarized collagen fiber was separated to hues by automated imaging software. The definition of the color categories orange-red or green-yellow was derived as follows: red, hue values 2–9 and 230–256; orange, 10–38; yellow, 39–51; and green, 52–128

(hue values) (Deguchi *et al*, 2005). The relative amount of each fiber color was used for orange-red (thick fiber) per yellow-green (thin fiber) ratio. For Fstl1 immunohistochemistry staining, antigen unmasking was performed by boiling slides in citric acid buffer. Avidin/biotin blocking was performed by using avidin/biotin blocking kit following manufacturer's protocol (Vector Laboratories). After tissue permeabilization by 0.1% TritonX-100 (American Bioanalytical), sections were blocked with 5% donkey serum for 30 min and then incubated with anti-mouse Fstl1 antibody at 1:100 dilution (polyclonal goat IgG, catalog #AF1738, R&D Systems) at 4°C overnight. After washing slides, sections were incubated with biotinylated anti-goat IgG for 30 min following HRP-polymer streptavidin antibody solution for 30 min and developed using DAB substrate kit (Vector Laboratories). α-SMA immunohistochemistry staining was performed using alkali phosphatase-conjugated α-SMA antibody at 1:50 dilution (clone 1A4, catalog #A5691, Sigma-Aldrich) and developed using alkaline phosphatase substrate kit (Vector® Red, Vector Laboratories). Hematoxylin staining (Sigma-Aldrich) was used for counter staining. The α-SMA-positive cell intensity was measured by Photoshop software. Pixel value of α-SMA-positive cells was normalized by infarcted myocardial area.

For immunofluorescent staining, heart samples were embedded in OCT compound (Sakura Finetek) and sectioned at 6 µm. Heart sections or cultured fibroblasts in slide chambers were fixed with 4% paraformaldehyde for 15 min. Permeabilization and serum blocking were performed as described earlier. The sections were incubated with primary antibody specific for mouse Fstl1 (polyclonal goat IgG, catalog #AF1738, R&D Systems, 1:50 dilution), sarcomeric actinin (monoclonal mouse IgG, clone EA-53, catalog #A7811, Sigma-Aldrich, 1:50 dilution), S100A4 (Rabbit, polyclonal, catalog# A5114, Dako cytomation, Denmark, 1:50 dilution), and α-SMA (clone 1A4, catalog #A2547, Sigma-Aldrich, 1:50 dilution) for overnight at 4°C. After washing, the sections were incubated with Alexa Fluor 488 or Alexa Fluor 594-conjugated secondary antibodies (Life Technologies). F-actin was stained by using Alexa Fluor 488-conjugated phalloidin antibody (Invitrogen). DAPI was used to detect nuclei. Fluorescent images were taken by confocal microscope LSM710 (Zeiss) or a BZ-9000 Keyence microscope. The details for each image are shown in Table EV1. Other antibodies used for investigation of cell source of Fstl1 are listed in Table EV2.

## qRT–PCR analysis

RNA from hearts or cultured cardiac fibroblasts was extracted using RNeasy Lipid Tissue Mini kit or RNeasy Micro Kit (both from Qiagen). Extracted RNA was reverse transcribed to complementary DNA by Superscript® III First-strand synthesis SuperMix (Life Technologies) according manufacture's protocol. qRT–PCR was performed using ViiA TM7 Real Time PCR system (Applied Biosystems) with Power SYBR® Green PCR Master Mix (Applied Biosystems). The relative levels of transcript was determined by using delta–delta CT methods and normalized by *GAPDH* and *36b4*. Primer sequences are listed in Table EV3.

## Statistical analysis

The results were presented as mean ± SEM. Statistical analysis was performed using GraphPad Prism 6 software (GraphPad Software)

## The paper explained

### Problem

Cardiac rupture followed by MI is life-threatening complication and frequently requires invasive surgical treatment. The heart secretes several hormones to maintain homeostasis and to adjust pathological stress conditions. Previously, we reported that Fstl1 protein is robustly upregulated following MI in a mouse model. However, the mechanistic role of Fstl1 induction in heart diseases is not fully understood.

### Results

In this study, we found that Fstl1 is robustly expressed in non-cardiomyocyte interstitial cells (fibroblasts and myofibroblasts) in the infarcted heart. The genetic ablation of Fstl1 in fibroblast lineage cells (Fstl1-cfKO) led to increased mortality after MI due to cardiac rupture. Fibroblasts and myofibroblasts play important roles in infarct healing by synthesizing matrix proteins, leading to an increase in cardiac integrity. In Fstl1-cfKO mice, hearts displayed decreased amounts of myofibroblasts and impaired matrix protein synthesis. Mechanistic studies showed that Fstl1 is essential for early cardiac fibroblast activation following MI and that these effects were mediated by the activation of an intracellular signaling-dependent mechanism.

### Impact

This study reveals that Fstl1 is essential for early fibroblasts activation that is required for acute repair of the infarcted heart. Thus Fstl1 could have utility as a therapeutic agent in the setting of MI.

and SPSS Statistics20 (IBM). Distribution of data (test of normality) was analyzed by *F*-test (GraphPad Prism) or Shapiro–Wilk test (SPSS 20). The data with normal distribution were assessed by parametric analysis: parametric *t*-test (unpaired, two-tailed) for two groups and ordinary one-way ANOVA and Tukey's multiple comparison tests for more than three groups. The data of non-normal distribution were assessed by nonparametric tests: Mann–Whitney *U*-test (two-tailed) for two groups and Kruskal–Wallis test and Dunnett's t3 test for multiple comparisons for more than three groups. Animal data of echocardiography, blood pressure, and qPCR were analyzed by two-way ANOVA, and *post hoc* multiple comparison was performed by Tukey's test. Survival curves were obtained by the Kaplan–Meier methods and compared by the log-rank test. $P < 0.05$ was considered as significant differences.

Expanded View for this article is available online.

## Acknowledgements

This work was supported by NIH grants HL081587, HL116591, HL120160, and HL126141 and Takeda Pharmaceuticals (K. Walsh, S. Maruyama). F.A. Recchia was supported by NIH grants HL74237 and HL108213. S. Maruyama was also supported by a Postdoctoral Research Fellowship in Cardiology by Japan Heart Foundation and Bayer (Japan). We thank Francesca Seta for consultation on the polarized microscopy. We are also grateful to Rouan Yao, Taina Rokotuiveikau, Matthew Phillippo, Maria Angeles Zuriaga Herrero, Reina Kobayashi, Miho Sano, and Blake Jardin for their assistance in mouse husbandry.

## Author contributions

SM, KN, KNP, SS, and IS performed MI surgery on mice. SM and KN performed histological staining of mice samples. SM and IS performed immunoblotting of mice samples and *in vitro* samples. SM, KN, and SS captured macro-images of mice organs and micro-image of sectioned samples with microscopes. SM isolated primary neonatal rat cardiac fibroblasts and performed *in vitro* assays. YA supervised production of recombinant Fstl1 protein. MJvdH provided Fstl1^flox/flox mice. SM and KW wrote the manuscript. NO and FAR provided scientific advice.

## Conflict of interest

The authors declared that they have no conflict of interest.

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
