## [Review Process File · EMBO Molecular Medicine]

Follistatin-like 1 promotes cardiac fibroblast activation and protects the heart from rupture

Sonomi Maruyama, Kazuto Nakamura, Kyriakos N. Papanicolaou, Soichi Sano, Ippei Shimizu, Yasuhide Asaumi, Maurice J. van den Hoff, Noriyuki Ouchi, Fabio A. Recchia, Kenneth Walsh

Corresponding author: Kenneth Walsh, Boston University School of Medicine

Review timeline:	Submission date:	15 December 2015
	Editorial Decision:	22 January 2016
	Revision received:	29 March 2016
	Editorial Decision:	20 April 2016
	Revision received:	27 April 2016
	Accepted:	27 April 2016

Transaction Report:

Editor: Roberto Buccione

1st Editorial Decision

22 January 2016

Thank you for the submission of your manuscript to EMBO Molecular Medicine. We have now heard back from the three Reviewers whom we asked to evaluate your manuscript.

Although the Reviewers agree on the potential interest of the manuscript, the issues raised are largely overlapping and of a fundamental nature and centre essentially on two main issues: 1) unresolved conflicts with published work, especially your own, and 2) insufficient support (and mechanistic insight) for the claims made.

I will not dwell into much detail, but I would like to highlight the main points from each Reviewer.

Reviewer 1 notes that based on previous knowledge, one would have expected no difference in cardiac rupture in the models reported in this manuscript. S/he also suggests that the important question (connected to the main novelty of your work) as to whether cardiac Fstl1 actions depend on amounts secreted or the cellular source, remains unresolved. This reviewer also lists other elements of discrepancy that would need to be dealt with experimentally.

Reviewer 2 asks for very important clarifications on the dataset and challenges the assumption made that Fst1 contributes directly to cardiac fibroblast differentiation, and as Reviewer 1, questions the causes for increased cardiac rupture in Fstl1 KO mice. S/he also asks for further experimentation to clarify a number of points, in partial overlap with Reviewer 1's comments..

Reviewer 3, similarly to #1, notes discrepancies with your previous work that should be reconciled. While the Reviewer suggests that this could be dealt with by further discussion, we tend to agree

with Reviewer 1 that further experimentation is required to specifically address the points. Reviewer 3 also notes, as do the other Reviewers, that a role for Fstl1 in maintaining cardiac integrity cannot be excluded. S/he also lists a number of other relevant points.

In conclusion, while publication of the paper cannot be considered at this stage, given the potential interest of your findings and after internal discussion, we have decided to give you the opportunity to address the criticisms.

We are thus prepared to consider a substantially revised submission, with the understanding that the Reviewers' concerns must be addressed with additional experimental data where appropriate and that acceptance of the manuscript will entail a second round of review. The overall aim is to significantly upgrade the relevance and conclusiveness of the dataset, which of course is of paramount importance for our title.

Please note that it is EMBO Molecular Medicine policy to allow a single round of revision only and that, therefore, acceptance or rejection of the manuscript will depend on the completeness of your responses included in the next, final version of the manuscript.

EMBO Molecular Medicine now requires a complete author checklist (<http://embomolmed.embopress.org/authorguide#editorial3>) to be submitted with all revised manuscripts. Provision of the author checklist is mandatory at revision stage; The checklist is designed to enhance and standardize reporting of key information in research papers and to support reanalysis and repetition of experiments by the community. The list covers key information for figure panels and captions and focuses on statistics, the reporting of reagents, animal models and human subject-derived data, as well as guidance to optimise data accessibility.

As you know, EMBO Molecular Medicine has a "scooping protection" policy, whereby similar findings that are published by others during review or revision are not a criterion for rejection. However, I do ask you to get in touch with us after three months if you have not completed your revision, to update us on the status. Please also contact us as soon as possible if similar work is published elsewhere.

I also suggest that you carefully adhere to our guidelines for publication in your next version, including our new requirements for supplemental data (see also below) to speed up the pre-acceptance process in case of a positive outcome.

***** Reviewer's comments *****

Referee #1 (Comments on Novelty/Model System):

This reviewer is concerned about the lack of specificity of the model chosen by the authors and the conclusion drawn by their findings for the following reason:

The Walsh group has extensively published on the biological and potential therapeutic role of Fstl1 in cardio-vascular protection. Based on their results, the group claimed several paracrine mechanisms for Fstl1 - importantly ! irrespective of its cellular source - to explain its pleiotropic actions:

- on cardiomyocytes Fstl1 seems to exert antihypertrophic and antiapoptotic effects
- on macrophages, Fstl1 apparently prevents activation of inflammatory gene expression
- on endothelial cells, Fstl1 seems to promote angiogenic processes
- on smooth muscle cells, Fstl1 inhibits proliferation.

In this paper, the authors now claim that cardiac fibroblasts are the major source for Fstl1 secretion in the ischemic heart. They therefore generated as conditional Fstl1-knock out mouse model to ablate Fstl1 in S100A4 expressing cell types in injured mouse heart. Using this model, they show a significant decrease of myocardial Fstl1 expression in response to myocardial infarction. The authors then causally link the Fstl1 decrease to the observed phenotype displaying increased cardiac

rupture and death.

They provide additional data showing that myocardial Fstl1 production isn't abrogated in cardiomyocyte-specific Fstl1 as well as in a myeloid cell/macrophage-specific Fstl1 knock out mice subjected to myocardial infarction.

Given previously published data by the Walsh group showing Fstl1 to exert its protective actions by a paracrine mechanism, one should expect unchanged rates of cardiac rupture in the two latter mouse models since Fstl1 effects are independent of its cellular source.

Such data are however not provided by the authors but they nevertheless claim specificity for the observed phenotype in their conditional S100A4/cardiac fibroblast Fstl1-knock out mouse model.

If the authors were able to show that cardiac rupture is actually unchanged in the cardiomyocyte- or macrophage-specific Fstl-1 knock out mouse model, their conclusion that cardiac fibroblasts are the most critical source would be clearly supported.

Their current data however do not answer the critical question whether cardiac Fstl1 actions actually depend on the dosage being secreted (irrespective of the cell type) or the cellular source.

Referee #1 (Remarks):

Maruyama and co-workers present novel and interesting findings pertaining to protective paracrine actions of Fstl1 in ischemic hearts of mice.

Using a conditional mouse model to ablate Fstl1 expression in S100A4 expressing cells in mice, they report an increase in cardiac rupture after experimental myocardial infarction (MI).

Use of this genetic model is driven by the author's hypothesis that cardiac fibroblasts (that express to some extent S100A4) are the major source of Fstl1 in the ischemic heart.

The authors went on demonstrating that cardiac fibroblasts are responsive to Fstl1 stimulation mediated by ERK1/2 activation.

Overall, they conclude from their results that Fstl1 being secreted from cardiac fibroblasts is indispensable for myocardial healing and scar formation.

Major Points:

The Walsh group has extensively published in the past on the biological and potential therapeutic role of Fstl1 in cardio-vascular protection.

Based on their results, the group as well as collaborators have claimed several numerous paracrine mechanisms and target cells for Fstl1 - irrespective of its cellular source - to explain the observed pleiotropic protective actions:

- i.e. on cardiomyocytes - Fstl1 seems to exert antihypertrophic and antiapoptotic effects (i.e. Ogura et al. Circ 2012).
- i.e. on macrophages - Fstl1 apparently prevents activation of inflammatory gene expression (i.e. Ogura et al. Circ 2012).
- on endothelial cells - Fstl1 seems to promote angiogenic processes (Ouchi et al. JBC 2008).
- on smooth muscle cells - Fstl1 inhibits proliferation (Miyabe et al. CVR 2014).

In this study, the authors now conclude that Fstl1 being secreted from cardiac fibroblasts (in response to ischemic damage of the heart) is the critical link to prevent cardiac rupture post-MI.

A) This reviewer is concerned that the data provided in the manuscript aren't sufficient to support this notion.

The authors provide additional data showing that myocardial Fstl1 production isn't abrogated in cardiomyocyte-specific Fstl1 as well as in a myeloid cell/macrophage-specific Fstl1 knock out mice subjected to MI.

Given previously published data by the Walsh group showing Fstl1 to exert its protective actions by paracrine mechanisms (also being secreted by cardiomyocytes), one should expect unchanged rates of cardiac rupture in the two latter mouse models since Fstl1 effects were reported to be independent

of its cellular source.

Such data are however not provided by the authors although they claim specificity for the observed phenotype in their conditional S100A4/cardiac fibroblast Fstl1-knock out mouse model.

The important questions thus remains unanswered whether Fstl-1 prevention of cardiac rupture is simply a matter of its dosage being secreted or its cellular source is a critical aspect (too), as claimed by the authors.

In addition, i.e. cardiomyocyte-specific Fstl1 KO mice treated with a neutralizing anti-Fstl1 antibody should display the same phenotype of enhanced cardiac rupture as the model used in this study.

Such result would also argue against cardiac fibroblasts as the necessary source of protective Fstl1 secretion and rather support a dose-dependent explanation for the observed phenotype.

This reviewer believes that these experiments are actually necessary to fully Support the authors conclusion drawn from their results.

B) in light of previously reported results by the Walsh group on paracrine actions of Fstl1 on above mentioned cardiac cell types, authors should be able to demonstrate that infarcted hearts display i.e.:

- i. increased inflammation since Fstl1 suppresses macrophage activation
- ii. enhanced post-MI cardiac remodeling since Fstl1 abrogates cardiomyocyte hypertrophy
- iii. diminished neoangiogenesis as Fstl1 improves EC-mediated angiogenesis
- iiii.

This analysis seems to be necessary to understand the relation of the currently used model of abrogated cardiac Fstl1 production with previously utilized models of ectopic (liver) or cardiomyocyte-derived Fstl1 generation.

C) The authors previously characterized several intracellular mechanism by which Fstl1 seems to exert its protective actions:

- i.e. activation of AMPK and inhibition of BMP-4 by Fstl1 in ischemic cardiomyocytes/cardiac tissue (Ogura et al. Circ 2012)
- i.e. stimulation of Akt1/2 (in addition to ERK1/2) and FOXO1/2 as well as mTor (Oshima et al. Circ 2008).
- i.e. stimulation of eNOS and GSK in endothelial cells (Ouchi et al. JBC 2008).

As a result, one should expect to find corresponding alterations in signaling pathways in the ischemic mouse hearts to corroborate the reported findings and understand them in light of previously reported cardiac actions of Fstl1.

Minor Points:

A) In Figure 1, increased cardiac Fstl1 expression is shown by Western blotting displaying multiple bands. Given the fact that this group and others (i.e. see most recent Fstl1 review by Chaly et al. Immunol Res 2014) highlights the need to discriminate between non-glycosylated and glycosylated forms of Fstl1 to appropriately decipher the biological actions, it seems necessary that authors provide an analysis of Fstl1 fractions being secreted from their model and i.e. the cardiomyocyte-specific Fst-1 KO mouse to assess whether different cardiac cell types might secrete similar or different Fstl1 fractions. This could also be important with respect to the authors claim that only cardiac fibroblast secreted Fstl1 fractions have the reported protective effect if they differs from those being released i.e. from cardiomyocytes.

B) In Table 1, authors report cardiac rupture in approximately 30% of permanently ligated/infarcted control mice. This is an unusually high percentage compared to other authors having previously published with this model. Could the authors comment on the difference ?

Summary: Overall, previously published data support the Notion that Fstl1 might play an important role in cardiac biology and could be of therapeutic interest for translational research. In light of the

extensive amount of data published, in particular by the Walsh group, this reviewer believes that specific claims made in this study must be further corroborated by additional experiments to strengthen the manuscript and allow the reader to align the novel findings with previous results.

Referee #2 (Remarks):

Summary: Previous studies have shown that Fstl1 (a member of SPARC family of proteins) is upregulated in response to cardiac injury, where it is largely thought to serve a protective role in the heart under stress. Systemic delivery of a recombinant form or transgenic expression of Fstl1 in the myocardium reduces cardiac myocyte apoptosis and inflammation after acute ischemia/reperfusion (I/R) injury. In addition, Ruiz-Lozano et al. recently reported that Fstl1 is strongly upregulated in cardiomyocytes and down-regulated in the epicardium following MI in mice. Reconstitution of local Fstl1 in the epicardium via collagen patches promotes the proliferation of immature cardiomyocytes and diminishes infarct size in mice, indicating that Fstl1 may also play part in cardiac repair. Herein the authors present data that formally add cardiac fibroblasts as the main source of the high levels of Fstl1 induction following MI in mice, further demonstrating that the partial ablation of Fstl1 in S100a4 cardiac fibroblast subpopulation (Fstl1-cfKO) prevents the proper healing of infarcted area, predisposing to cardiac rupture. The study addresses an important topic with appropriate genetic models, cell biology and biochemical tools. However, a number of clarifications are needed for this otherwise interesting paper.

Specific

- 1) Genetically engineered mouse models provided excellent data to support the contention that cardiac fibroblasts are the main source of the surge of Fstl1 in the infarcted tissue and serum in MI mice. S100a4cre[±] x Fstl1 flox/flox mice show marked reduction in the Fstl1 following myocardial infarction (also in cardiac fibroblasts harvested in the neonatal period), but not MHCcre[±] x Fstl1 flox/flox and LyzMcrc[±] x Fstl1 flox/flox mice, excluding the contribution of cardiomyocytes or myeloid-derived cells/macrophages to the upregulation of Fstl1 in infarcted mice. Observations coherent with these postulates were also provided by extensive immunohistochemical data, which showed the co-localization of Fstl1 staining with S100a4 and alpha-SMA, a marker of active myofibroblasts. Gain and -loss of function experiments in cardiac fibroblasts confirmed the influence of Fstl1 on the ability of cardiac fibroblasts to migrate and to proliferate via activation of Erk1/2. However, Fstl1 does not seem to contribute directly to differentiation of cardiac fibroblasts into myofibroblasts, despite the fact that a reduced number of myofibroblasts (alpha-SMA staining) was found in the infarcted area of Fstl1-cfKO compared to WT mice. It could be the case that the lack of Fstl1 predisposes to cardiac fibroblast apoptosis, with mechanistic implications for the increased rate of cardiac rupture in Fstl1-cfKO.
- 2) Graphs in Figure 1A and B show marked average increases of Fstl1 protein in the ischemic area and in the serum of three (3) mice subjected to LAD coronary artery ligation, peaking at 7 days after surgery. Data in Figure 3B and C (14-16 mice) seem to indicate some variability in the increases of Fstl1 following MI in mice. Given the demonstration here that the increases in Fstl1 are due to cardiac fibroblasts it would be important to examine whether there is a quantitative correlation between the serum and tissue levels of Fstl1 and the extension of infarcted area, as well as the number of alpha-SMA positive cells.
- 3) Please clarify whether mice used in the experimental groups shown in Figure 3 were also used to generate data shown in Figure 4. If a total of 59 MI-WT and 30 MI-Fstl1-cfKO mice were used to examine the mortality after myocardial infarction why only 15 WT and 14 Fstl1-cfKO were used in echocardiographic analysis (6 days post-MI) and the same numbers were euthanized (7 days post-MI) for analysis? A complete set of echocardiographic and histologic data, including those from WT and Fstl1-cfKO mice who survived after 7 days post-MI should be included.
- 4) Risk factors that predispose to cardiac rupture in clinics include large myocardial infarction and evidence of infarct expansion or extension. Given the recent demonstration that Fstl1 might contribute to cardiac repair, the authors should consider the possibility of a combined effect of

defects in the fibrogenesis and in cardiomyocyte proliferation as important factors explaining the predisposition to cardiac rupture in Fstl1-cfKO mice.

Referee #3 (Remarks):

The study by Maruyama et al. investigates the significance of follistatin-like 1 (Fstl1) produced by cardiac fibroblasts during the response of the heart to myocardial infarction. The authors demonstrate that Fstl1, whose expression is induced post infarction, originates primarily from cardiac fibroblasts. Then, taking advantage of fibroblast-specific Fstl1 knockouts, they show that Fstl1 deletion leads to an increased occurrence of cardiac rupture following myocardial infarction. Data suggest that migration and proliferation are affected by the absence of Fstl1 expression in activated cardiac fibroblasts, which results in a decreased extracellular matrix production that affects the healing capacity of the heart.

The importance of Fstl1 in cardiac homeostasis has been already studied in different animal models of cardiac disease, including by the authors themselves. Fstl1 is known to be protective in the stressed heart. However, the cellular origin of Fstl1 appears to depend on the type of injury and on the phase of the response. The originality of the paper is therefore in the attempt to investigate whether cardiac fibroblasts represent a significant source of Fstl1 during the acute phase of myocardial infarction, which contributes to cardiac repair. In this regard, the described experiments are rather convincing. The main problem is to reconcile the information available from studies previously published by the authors to create a uniform model. Some discrepancies emerge from these different studies that should be addressed. The paper by Wei et al. (Nature; 2015) should be more extensively discussed since this work takes advantage of the same myocardial infarction model used in the present study but reaches a different conclusion on the origin and the effect of Fstl1 in the heart.

Major points

1. Fstl1 is subjected to post-translational modification that affects its activity. Figure 1A depicts at least three isoforms, Figure 1B possibly two, and Figure 3 only one. What is the isoform that is produced by cardiac S100A-positive fibroblasts? Could it be that fibroblasts produce an isoform characterized by less glycosylation? It seems that the 37KD isoform, which is predominantly expressed in the heart post infarction, is consistent with the hypoglycosylated isoform produced by the epicardium.
2. Along the same lines, the Fstl1 isoforms that are detected in cell lysates (non-secreted) vs. media (secreted) are different (Figure 3)? The apparent molecular weight of the non-secreted form corresponds to the isoform produced in the injured heart. Which one do the authors consider important for the response to injury? Does Fstl1 produce primarily autocrine or paracrine effects?
3. In cfKO, one cannot rule out the contribution of epicardial cells as a source of Fstl1. Co-immunostaining with antibodies directed against Fstl1 and Wt1 would be informative.
4. In their previous work (Wei et al. Nature. 2015), the authors described massive cardiomyocyte-specific Fstl1 staining in the heart 7-14 days post infarction. Why is this not apparent in the present study?
5. The physiological data described in Figure 4 and in Table 1 and 2 are convincing. However, the authors focus on the acute phase. Although function is not different in the two genotypes 7 days post MI, it remains possible that Fstl1 plays a role in maintaining cardiac integrity during the chronic period. Do the authors have investigated cardiac dimensions and function during the chronic phase?
6. Figure EV 4D suggests that Fstl1 ablation has no impact on fibronectin expression but this is in contrast to what shown in Figure 5B.

Minor points

1. It would be nice to have control sham hearts for comparison in Figure 1A and B
2. Contrary to what stated in the text (page 6), Figure 2D does not support expression of Fstl1 in aSMA-positive cells.
3. What is the level of S100A expression in cfKO post MI (Figure 3)?

4. Data in Figure 5C should be quantified, and normalized per heart section.
5. Data in Figure 7A should be quantified.

1st Revision - authors' response

29 March 2016

We are deeply appreciative of the comments from the three reviewers. In many cases we performed the requested experiments. In a few instances we clarified the text after realizing that confusion was caused by some of the statements made in the original version of our manuscript.

We wish to emphasize that the main novel findings of our work are the following:

- 1) Multiple lines of evidence are provided to show that cardiac fibroblasts are a major source, and most likely the predominant source, of Fstl1 production in the injured heart.
- 2) A new line of mice is created and analyzed where Fstl1 is partially ablated in fibroblasts.
- 3) The major phenotype of fibroblast-specific Fstl1 deficiency is cardiac rupture (and mortality) in the myocardial infarction model.
- 4) Mechanistically, it is shown that Fstl1 does not directly affect myofibroblast differentiation, but it functions at an earlier stage of fibroblast activation, promoting the proliferation and migration of this cell type.

We believe that our work is appropriate for *EMBO Molecular Medicine* because cardiac rupture is a significant, life-threatening condition associated with myocardial infarction. Our work suggests that Fstl1 could have therapeutic utility as a novel regulator of cardiac matrix and the early healing process in the heart.

Referee #1:

The Walsh group has extensively published on the biological and potential therapeutic role of Fstl1 in cardio-vascular protection.

Response: Thank you for this comment. Our lab was the first to publish on the role of Fstl1 in cardiovascular protection (Oshima et al. *Circulation* 2008) and this has now been the topic of 20 publications from at least 8 laboratories. While some secreted factors are much more widely studied in the cardiovascular system (e.g. compared to thousands of citations for VEGF, BNP or AngII), we believe that these early experimental and clinical data make a convincing case that Fstl1 is an important regulator of cardiovascular pathology.

What is novel here is the first use of murine genetic models to document an effect of Fstl1 in experimental myocardial infarction. Using this model we show, for the first time, that cardiac fibroblasts are the predominant source of Fstl1 in the post-MI heart and that Fstl1's expression by these cells is essential for early cardiac fibroblast activation and prevention of cardiac rupture post-MI.

They provide additional data showing that myocardial Fstl1 production isn't abrogated in cardiomyocyte-specific Fstl1 as well as in a myeloid cell/macrophage-specific Fstl1 knock out mice subjected to myocardial infarction. Given previously published data by the Walsh group showing Fstl1 to exert its protective actions by a paracrine mechanism, one should expect unchanged rates of cardiac rupture in the two latter mouse models since Fstl1 effects are independent of its cellular source.

Such data are however not provided by the authors but they nevertheless claim specificity for the observed phenotype in their conditional S100A4/cardiac fibroblast Fstl1-knock out mouse model. If the authors were able to show that cardiac rupture is actually unchanged in the cardiomyocyte- or macrophage-specific Fstl1 knock out mouse model, their conclusion that cardiac fibroblasts are the most critical source would be clearly supported.

Their current data however do not answer the critical question whether cardiac Fstl1 actions actually depend on the dosage being secreted (irrespective of the cell type) or the cellular source.

Response: We agree that it is most likely that Fstl1 exerts its protective actions via a paracrine (or autocrine) mechanism. The actions of Fstl1 have been shown to be mediated by a cell surface receptor, and its binding to this receptor is dose-dependent and saturable (see, for example, Ouchi et

al. *J. Biol. Chem.* 2010). We document that Fstl1 levels are reduced in the hearts of the S100a4^{cre/+} x Fstl1^{fllox/fllox} mice, leading to a reduction in cardiac fibroblast activation and cardiac rupture. In contrast, there is no detectable contribution of cardiac myocyte or macrophage Fstl1 to the high levels of expression that are observed in the post-MI heart. While our findings are novel, they are also straightforward and consistent with basic biochemical principles of protein dose and receptor occupancy. While more convoluted hypotheses may exist to explain the cardiac rupture phenotype, we believe that the most logical explanation is that the cardiac fibroblast is a functionally and quantitatively significant source of Fstl1 in the post-MI heart. We have rewritten the section in Discussion to clarify these points.

In this regard, we believe that the cardiomyocyte- or the macrophage-specific Fstl1 mice would display no detectable phenotype in the MI model because these genetic manipulations do not decrease the levels of this factor in the heart under these conditions. Approximately 120 mice were used to document the “positive” result with the cardiac fibroblast KO mice. Thus, subjecting myocyte-KO and macrophage-KO mice to surgeries would not be a wise use of limited resources and may represent an ethically questionable use of laboratory animals. We further discuss this issue in the response to point 1A below.

A) This reviewer is concerned that the data provided in the manuscript aren't sufficient to support this notion. The authors provide additional data showing that myocardial Fstl1 production isn't abrogated in cardiomyocyte-specific Fstl1 as well as in a myeloid cell/macrophage-specific Fstl1 knock out mice subjected to MI. Given previously published data by the Walsh group showing Fstl1 to exert its protective actions by paracrine mechanisms (also being secreted by cardiomyocytes), one should expect unchanged rates of cardiac rupture in the two latter mouse models since Fstl1 effects were reported to be independent of its cellular source. Such data are however not provided by the authors although they claim specificity for the observed phenotype in their conditional S100A4/cardiac fibroblast Fstl1-knock out mouse model.

The important questions thus remains unanswered whether Fstl-1 prevention of cardiac rupture is simply a matter of its dosage being secreted or its cellular source is a critical aspect (too), as claimed by the authors. In addition, i.e. cardiomyocyte-specific Fstl1 KO mice treated with a neutralizing anti-Fstl1 antibody should display the same phenotype of enhanced cardiac rupture as the model used in this study. Such result would also argue against cardiac fibroblasts as the necessary source of protective Fstl1 secretion and rather support a dose-dependent explanation for the observed phenotype.

Response: The point that the reviewer makes is that cardiac myocytes (and potentially other cells) make Fstl1, and that Fstl1 from these other sources should compensate for the loss of Fstl1 from the fibroblast knockout (i.e. the S100a4^{cre/+} x Fstl1^{fllox/fllox}). It is true that Fstl1 is made by multiple cell types, but few studies have examined the relative contributions from the different cell sources that are responsible for its upregulation in response to injury. To rigorously address this issue we have been employing mouse genetic approaches.

As the reviewer correctly points out, Fstl1 is produced by cardiac myocytes but, as shown in the manuscript, it does not contribute to a quantitatively significant level of overall cardiac Fstl1 in the heart under the conditions of MI injury. The reasons for this are: 1) the post-infarct scar is largely devoid of cardiac myocytes; 2) on a cell-by-cell basis, activated cardiac fibroblasts make more Fstl1 than cardiac myocytes and 3) in post-MI hearts there is massive fibroblast activation and these are the cells that primarily repopulate the infarcted region.

To explain this issue in greater detail, we bring the reviewer's attention to the only other paper that has addressed the cell sources of Fstl1 in the heart (Shimano *Proc. Natl. Acad. Sci. USA* 2011). As shown in this study, transverse aortic constriction (TAC) leads to Fstl1 induction (but far below the level that is observed following permanent LAD ligation):

Notably, when *Fstl1* is ablated in cardiac myocytes using α -myosin heavy-chain-Cre recombinase, *Fstl1* transcript and protein expression are reduced approximately 40-46% at baseline and in response to TAC (see Figures A and B from Shimano *Proc. Natl. Acad. Sci. USA* 2011). Thus, more than half of *Fstl1* expression is derived from the non-myocyte compartment at baseline and in the TAC model. To examine the cell type-specific expression of *Fstl1* in greater detail, tissue sections of WT and cardiac myocyte-specific *Fstl1*-KO hearts were immunostained for *Fstl1* and the cardiac myocyte marker α -actinin and analyzed by confocal microscopy (Fig. F from Shimano et al.). Dual-image analysis of *Fstl1* (green) and α -actinin (red) showed a colocalization signal (yellow) in WT mice. No colocalization was detected in *Fstl1*-KO heart sections. Notably, an intense *Fstl1* signal (green) could be observed in the relatively rare non-myocyte cells within these sections, but the identity of the intensely *Fstl1*-positive, non-myocyte cells was a mystery.

In the current study we provide multiple lines of evidence to show that: 1) the non-myocyte source of *Fstl1* in the heart is a fibroblast; 2) the fibroblast is a predominant source of *Fstl1* in the post-MI heart and 3) cardiac fibroblast *Fstl1* is essential to prevent cardiac rupture. As mentioned previously, it is important to note that the contribution of cardiac myocytes to overall *Fstl1* production is negligible in the post-MI heart due to the very high level of fibroblast activation (in fibroblasts that express high levels of *Fstl1*), and due to the dropout of myocytes (that express relatively low levels of *Fstl1*).

*B) in light of previously reported results by the Walsh group on paracrine actions of *Fstl1* on above mentioned cardiac cell types, authors should be able to demonstrate that infarcted hearts display i.e.:*

- i. increased inflammation since *Fstl1* suppresses macrophage activation*
- ii. enhanced post-MI cardiac remodeling since *Fstl1* abrogates cardiomyocyte hypertrophy*
- iii. diminished neoangiogenesis as *Fstl1* improves EC-mediated angiogenesis*

Response: In response to Reviewer point i, we measured macrophage infiltration in the post-MI heart. We found that there was no statistically significant difference between the numbers of F4/80 positive cells in the infarct area between the WT and the cfKO hearts post-MI. These data are shown below and are included in the Supplement of the revised manuscript (Figure EV 5B). While our colleagues have previously published that *Fstl1* is anti-inflammatory in some contexts, we assume that the pro-inflammatory signaling in the post-infarct heart overwhelms any contribution of *Fstl1* or that *Fstl1* is absent in the post-MI heart.

Detection of macrophages by F4/80 staining (AbD Serotec, Clone A3-1) in the ischemic myocardium of WT and cfKO mice at day 7 after MI. DAB substrate was used for detection. Counter staining was performed using hematoxylin stain. F4/80 positive cell density in the infarcted area was measured as pixel at high magnification. Scale bar indicates 100 μ m. Error bars represent mean \pm SEM (n=12 and 14 for WT and cfKO, respectively). Statistical analysis was performed by unpaired t-test (two-tailed).

In response to Reviewer point ii, we measured cardiac myocyte cross-sectional area in the remote zone (where myocytes remain intact) and found no statistically significant differences between WT and cfKO strains at baseline or post-MI. These data are shown below and are included in the data supplement of the revised manuscript (Fig EV 8). While an effect of *Fstl1* ablation on cardiac hypertrophy is observed in the TAC/pressure overload model, which is a less complex model of hypertrophy (Shimano *Proc. Natl. Acad. Sci. USA* 2011), we assume that the pro-

hypertrophic signaling in the post-MI heart overwhelms any contribution of Fstl1 or Fstl1 is absent under these conditions.

Cardiomyocyte hypertrophy was assessed by measuring cardiomyocyte cross-sectional area of non-ischemic myocardium at 7 days after MI. The paraffin section was stained using FITC-conjugated GWA antibody (Life Technologies) to detect the outline of the cardiomyocytes. Error bars represent mean ± SEM (n=150 cardiomyocytes for each group). Statistical analysis was performed by Kruskal-Wallis and Dunn's test for post hoc analysis.

In response to Review point iii, we assessed capillary density in the cardiac remote zone under different experimental conditions. We found that the cfKO displayed a lower capillary density than WT hearts in the post-MI condition. These data are shown below and they are included in the Supplement of the revised manuscript (Fig. EV 7). These data are consistent with our previous finding that Fstl1 promotes endothelial cell function and thereby facilitates an angiogenic response in the hind limb ischemic model (Ouchi et al. *J. Biol. Chem.* 2008). However, to the best of our knowledge, there is no known role for the microvasculature in post-MI cardiac rupture.

Capillary density was determined by staining with Alexa-594 conjugated Isolectin-IB4 antibody (Life Technologies) in WT and cfKO heart at day 7 after sham and MI surgery. DAPI was used for nuclei staining. Capillary number per cardiomyocyte was measured in the border zone of infarcted heart and the left ventricle of sham heart. Error bars represent mean ± SEM (n=150 cardiomyocytes for each group). Statistical analysis was performed by Kruskal-Wallis test and Dunn's test for post hoc analysis.

C) The authors previously characterized several intracellular mechanism by which Fstl1 seems to exert its protective actions...AMPK; Akt1/2; eNOS and GSK... As a result, one should expect to find corresponding alterations in signaling pathways in the ischemic mouse hearts to corroborate the reported findings and understand them in light of previously reported cardiac actions of Fstl1.

Response: We analyzed AMPK and Akt phosphorylation/activation status, that are situated upstream of eNOS, GSK3beta and Foxo, in the remote zone of the different experimental groups of mice. The deficiency of Fstl1 led to reductions of both Akt and AMPK in the post-MI heart. These findings are consistent with previously published data showing that Fstl1 is upstream of both of these signaling pathways in other contexts (e.g. Oshima et al. *Circulation* 2008 and Ogura et al. *Circulation* 2012). These data are shown below and they are included in the data Supplement of the manuscript (Fig EV 9).

Detection of AMPK and Akt signals in the heart at day 7 after sham or MI surgery. All antibodies were purchased from Cell Signaling: p-AMPKα (T172) antibody (Cat. #2535), AMPKα antibody (Cat. #2532), p-Akt (Ser473) antibody (Cat. #4058), Akt antibody (Cat. #9272) and tubulin antibody (Cat. #2148). Error bars represent mean ± SEM (n=3 for each group). Statistical analysis was performed by

two-way ANOVA and Tukey's test for post hoc analysis.

Minor Points:

A) In Figure 1, increased cardiac *Fstl1* expression is shown by Western blotting displaying multiple bands. Given the fact that this group and others (i.e. see most recent *Fstl1* review by Chaly et al. *Immunol Res* 2014) highlights the need to discriminate between non-glycosylated and glycosylated forms of *Fstl1* to appropriately decipher the biological actions, it seems necessary that authors provide an analysis of *Fstl1* fractions being secreted.

Response: We appreciate the Reviewer's point. When *Fstl1* is blotted in the whole heart it represents the composite of its differentially glycosylated forms that are secreted from cardiac fibroblasts and cardiac myocytes. This is illustrated in the figure panel below on the left, and this figure is also included in the data Supplement of the revised manuscript (Fig. EV 12). As can be seen, both cardiac fibroblasts (FB) and cardiac myocytes (CM) secrete glycosylated forms of *Fstl1* but the material from the cardiac myocyte fraction appears to be slightly more glycosylated due to its reduced electrophoretic mobility (red arrows). Notably, the intracellular form (in the cell lysate) appears to be much less glycosylated than the secreted form (in the media), as we reported previously (Oshima et al. *Circulation* 2008). Treatment with tunicamycin, to block glycosylation, led to a form of *Fstl1* with relatively high electrophoretic mobility. These data are highly similar to those reported in Fig. 4g by Wei et al. (*Nature* 2015) that is shown in the panel on the right below.

While this provides an explanation for the multiple *Fstl1* bands in the gels, we point out that we detect no functional differences of any of the glycosylated forms (or the non-glycosylated form) in their ability to activate fibroblasts (discussed in response to Reviewer 3, comment #2).

changed from FBS 10% contained DMEM/F-12 to 0% FBS for CM and 0.5% FBS. Cells were cultured with or without tunicamycin (1 μ g/ml) for 16 hours. Conditioned media was concentrated by Amicon Ultra filter 10k device (14,000xg, 10min). Mouse *Fstl1* protein was detected by western blotting.

B) In Table 1, authors report cardiac rupture in approximately 30% of permanently ligated/infarcted control mice. This is an unusually high percentage compared to other authors having previously published with this model. Could the authors comment on the difference?

Response: The level of cardiac rupture observed in our experiments is as expected from the definitive study of van den Borne et al. *Cardiovasc Res.* 2009. Figure 2A from this paper is shown below for the Reviewer's reference. Our mice are a cross between C57Bl6 and BalbC. The consistency between our work and that of van den Borne et al. is noted in the revised manuscript.

(van den Bourne *Cardiovasc Res.* 2009)

Referee #2:

Ruiz-Lozano et al. recently reported that Fstl1 is strongly upregulated in cardiomyocytes and down-regulated in the epicardium following MI in mice.

Response: We respectfully disagree with this statement. Wei et al. *Nature* 2015 did not report that Fstl1 is strongly upregulated in cardiomyocytes. They reported the large upregulation of “myocardial”, not cardiac myocyte, Fstl1 in the post-MI heart, but they did not identify the cell source. As can be seen by the immunofluorescence in Figure 2I from Wei et al. (shown below), there is little or no overlap of Fstl1 (red) with the cardiac myocyte marker Tnni3 (green). Furthermore, the immunohistochemistry in Figure 2i of Wei et al. appears to us to be very consistent with expression by cardiac fibroblasts, but identification of the cell source was not addressed in their study.

In contrast, our current study provides multiple lines of evidence to show that fibroblasts are a significant source of cardiac Fstl1 and that the genetic deficiency of Fstl1 in these cells produces a cardiac rupture phenotype.

Fig. 2i**Fig 2I**Wei *Nature* 2015

The study addresses an important topic with appropriate genetic models, cell biology and biochemical tools. However, a number of clarifications are needed for this otherwise interesting paper.

Specific

1) Genetically engineered mouse models provided excellent data to support the contention that cardiac fibroblasts are the main source of the surge of Fstl1 in the infarcted tissue and serum in MI mice. S100a4cre^{+/-} X Fstl1flox/flox mice show marked reduction in the Fstl1 following myocardial infarction (also in cardiac fibroblasts harvested in the neonatal period), but not MHCcre^{+/-} x Fstl1flox/flox and LyzMcrc^{+/-} x Fstl1flox/flox mice, excluding the contribution of cardiomyocytes or myeloid-derived cells/macrophages to the upregulation of Fstl1 in infarcted mice. Observations coherent with these postulates were also provided by extensive immunohistochemical data, which showed the co-localization of Fstl1 staining with S100a4 and alpha-SMA, a marker of active myofibroblasts. Gain and -loss of function experiments in cardiac fibroblasts confirmed the influence of Fstl1 on the ability of cardiac fibroblasts to migrate and to proliferate via activation of Erk1/2. However, Fstl1 does not seem to contribute directly to differentiation of cardiac fibroblasts into myofibroblasts, despite the fact that a reduced number of myofibroblasts (alpha-SMA staining) was found in the infarcted area of Fstl1-cfKO compared to WT mice.

Response: Yes; it is little complicated, but this is the crux of the argument. Fstl1 clearly does not have a direct effect on fibronectin expression (α SMA, etc.) based upon mechanistic cell culture experiments (shown in Fig. EV 11). In contrast, Fstl1 manipulations lead to changes in the fibroblast proliferation and migration in the cell culture experiments (Figures 7 and 8). However, the deficiency of Fstl1 in vivo leads to reductions in fibronectin (α SMA, etc.) expression. Thus, we conclude that the role of Fstl1 in the heart is the early activation of fibroblasts (proliferation and migration) and that the effects of Fstl1-deficiency on myofibroblast differentiation in vivo are an indirect, or secondary, effect of having reduced fibroblast activation/number in this context. This model is elaborated in the Discussion and shown in Figure 8D. Overall, we strongly believe that our work is of high priority because it represents the most definitive understanding of how Fstl1 contributes to a fibrotic response in any context.

It could be the case that the lack of Fstl1 predisposes to cardiac fibroblast apoptosis, with mechanistic implications for the increased rate of cardiac rupture in Fstl1-cfKO.

Response: Thank you for this important point. To address this question we stained fibroblasts for vimentin and TUNEL to assess for differences in the frequency of apoptotic cell death in the different strains. There were no differences in the frequencies of TUNEL/vimentin double-positive cells between the wild-type and cfKO strains. These new data are shown below and they are included in the data Supplement of the revised manuscript (Figure EV 6).

Fibroblast apoptosis was assessed by TUNEL and vimentin co-staining. WT and cfKO hearts sampled at day 7 after MI surgery were used. DAPI was used for nuclei staining. TUNEL staining was performed using FITC-conjugated *in Situ* Cell Death Detection Kit (Roche) following the manufacturer's protocol. Vimentin was detected using anti-vimentin antibody (Santa Cruz Biotechnology, H-84) and secondary donkey anti-rabbit AF598 conjugated antibody (Life Technologies). DAPI was used for nuclei staining. TUNEL and vimentin double positive cells in the infarcted area and border zone were counted at high magnification. Error bars represent mean ± SEM (n=12 for WT and cfKO). Statistical analysis was performed by unpaired t-test (two-tailed).

2) Given the demonstration here that the increases in Fstl1 are due to cardiac fibroblasts it would be important to examine whether there is a quantitative correlation between the serum and tissue levels of Fstl1 and the extension of infarcted area, as well as the number of alpha-SMA positive cells.

Response: Thank you for this comment. Using stored samples, we analyzed serum cardiac myosin light chain-1 (CMLC-1) and compared this value to cardiac (A) and serum (B) Fstl1 levels at 7 days post-MI. Statistically significant correlations between these parameters were observed (Fig. EV 1). Furthermore, we found a statistically significant correlation between the expression of cardiac α -smooth muscle actin and Fstl1 (C). These data are shown below and included in the data Supplement of the revised manuscript (Fig. EV 2). These data are consistent with the hypothesis that the level Fstl1 induction is dependent upon the degree of cardiac injury and the degree of cardiac fibroblast activation.

The relationship between tissue Fstl1 level and α SMA expression was assessed. All samples were harvested at day 7 after MI. Ischemic myocardium of WT mice heart was processed as described in the main manuscript. Mouse Fstl1 and α SMA protein were detected by western blotting. Correlation was assessed by GraphPad Prism 6 software (n=14).

3) Please clarify whether mice used in the experimental groups shown in Figure 3 were also used to generate data shown in Figure 4.

Response: Yes. Surviving mice at day 7 were utilized for the protein and transcript analyses in Fig. 3. This point was clarified in the Result section of the revised manuscript.

4) *Fstl1* might contribute to cardiac repair, the authors should consider the possibility of a combined effect of defects in the fibrogenesis and in cardiomyocyte proliferation as important factors explaining the predisposition to cardiac rupture in *Fstl1*-cfKO mice.

Response: According to the report by Wei et al. (*Nature* 2015) the level of cardiac myocyte proliferation in the post-MI heart is 0.5 cells/mm², based upon pH3⁺ and Aurora B⁺ cell frequency, and this level would be predicted to be lower in *Fstl1*-deficient mice. (In the patch plus *Fstl1* condition, the level of myocyte proliferation increased to approximately 1 to 2 cells/mm² based upon the measurement.)

In contrast, we calculate that the fibroblast level is 3700 cells/mm² by assessing the frequency of vimentin⁺ cells in the infarct area. In light of the ~7,000-fold higher level of fibroblasts than proliferating myocytes, and the widely recognized role of fibroblasts in the production of cardiac matrix, it is difficult to imagine a quantitatively significant role of reduced myocyte proliferation in cardiac rupture. Furthermore, we focused on the cardiac fibroblast in this study because this cell type is recognized to produce the matrix that determines the frequency of post-MI cardiac rupture.

Referee #3:

The originality of the paper is therefore in the attempt to investigate whether cardiac fibroblasts represent a significant source of Fstl1 during the acute phase of myocardial infarction, which contributes to cardiac repair. In this regard, the described experiments are rather convincing. The main problem is to reconcile the information available from studies previously published by the authors to create a uniform model. Some discrepancies emerge from these different studies that should be addressed. The paper by Wei et al. (Nature; 2015) should be more extensively discussed since this work takes advantage of the same myocardial infarction model used in the present study but reaches a different conclusion on the origin and the effect of Fstl1 in the heart.

Response: Thank you for this comment. However, we take issue with the statement, “*this work [Wei et al. Nature; 2015] takes advantage of the same myocardial infarction model used in the present study but reaches a different conclusion on the origin and the effect of Fstl1 in the heart*”. While it is true that both studies examine the MI model, the reviewer should note that Wei et al. *Nature* 2015 reported that the delivery of a hypoglycosylated form of *Fstl1* (produced in bacteria, source: Aviscera Bioscience) via an epicardial patch stimulates the proliferation of pre-existing myocytes. It was also reported that transgenic expression of *Fstl1* did not have this activity (this latter set of experiments were performed in the Walsh laboratory). Wei et al. also reported that the expression of epicardial *Fstl1* is silenced in the post-MI heart. Nothing in our current paper challenges these findings.

No genetic loss-of-function studies were conducted by Wei et al., whereas mouse models of genetic deficiency are the methodology employed in our current study. Using the KO mouse approach, we observed that the fibroblast is a significant source cardiac *Fstl1* and the deficiency of *Fstl1* from this cell source leads to rapid cardiac rupture, the predominant phenotype. In mechanistic studies, we focus on the fibroblast because the matrix produced by this cell type is widely appreciated to be a critical determinant of cardiac rupture. Furthermore, the frequency of fibroblasts that repopulate the infarct are ~3,600 cells mm² that is in vast excess of myocyte proliferation (reported by Wei et al. to be 0.5 proliferative myocytes/mm² in WT mice).

We clarify these issues in our revised manuscript and more extensively discuss the study of Wei et al. *Nature* 2015.

Major points

1. What is the isoform that is produced by cardiac S100A-positive fibroblasts? Could it be that fibroblasts produce an isoform characterized by less glycosylation?

Response: The Fstl1 isoform produced in cardiac fibroblasts is highly glycosylated compared to Fstl1 produced in bacteria (or when cells are treated with tunicamycin).

As discussed in response to a point made by Reviewer 1, when Fstl1 is blotted in the whole heart it represents the composite of its differentially glycosylated forms that are secreted from cardiac fibroblasts and cardiac myocytes. This is illustrated in figure panel B below, and this figure is also included in the data Supplement of the revised manuscript (Fig. EV 12A). As can be seen, both cardiac fibroblasts (FB) and cardiac myocytes (CM) secrete glycosylated forms of Fstl1 but the material from the cardiac myocyte fraction appears to be slightly more glycosylated due to its reduced electrophoretic mobility (red arrows). Notably, the intracellular form (in the cell lysate) appears to be less glycosylated than the secreted form (in the media), as we reported previously (Oshima et al. *Circulation* 2008) and shown below in panel A. Treatment with tunicamycin, to block glycosylation, led to a form of Fstl1 with relatively high electrophoretic mobility. These data are highly similar to that reported in Fig. 4g by Wei et al. (*Nature*, 2015) that is shown in the panel on the right below.

While this provides an explanation for the multiple Fstl1 bands in the gels, we point out that we detect no differences in any of the glycosylated forms in their ability to activate fibroblasts (see response to point 2 below).

2. Along the same lines, the Fstl1 isoforms that are detected in cell lysates (non-secreted) vs. media (secreted) are different (Figure 3)? The apparent molecular weight of the non-secreted form corresponds to the isoform produced in the injured heart. Which one do the authors consider important for the response to injury? Does Fstl1 produce primarily autocrine or paracrine effects?

Response: Over the years, my laboratory has produced and evaluated a number of different Fstl1 formulations. Some of these formulations were prepared by the Walsh lab or by a contract research organization for the Walsh lab. A partial list of these formulations is shown in the Table below. Based upon electrophoretic mobility (A), the degree of Fstl1 glycosylation can be ranked as follows: mammalian cell-produced > insect cell-produced > bacterial cell produced; as would be expected.

Importantly, we do not find that the bioactivity Fstl1 differs with respect to its level of glycosylation when evaluating its ability to activate cardiac fibroblasts. A representative experiment is shown below (B). These data are also included in the Supplemental data section of the revised manuscript (Fig EV 12 and Table EV V).

(A) Western blot analysis of mouse Fstl1 protein in cell lysates and secreted from cardiomyocytes and cardiac fibroblasts. Neonatal rat cardiomyocytes and cardiac fibroblasts were infected with adenovirus encoding mouse Fstl1 (50 MOI) for 24 hours. Culture media was changed from FBS 10% contained DMEM/F-12 to 0% FBS for CM and 0.5% FBS. Cells were cultured with or without tunicamycin (1µg/ml) for 16 hours. Conditioned media was concentrated by Amicon Ultra filter 10k device (14,000xg, 10min). Mouse Fstl1 protein was detected by western blotting. (B) Molecular size of multiple Fstl1 recombinant proteins was assessed by western blotting. Detailed information for each recombinant protein is listed in Table EV V. Equal amount of proteins (5ng/lane) were loaded to 4-12% TGX gel and transferred to PVDF membrane. Fstl1 proteins were detected using human and mouse Fstl1 polyclonal antibody (both from R&D Systems).

With regard to your comment about a “paracrine” vs. an “autocrine” action, this is a very difficult issue to address experimentally for any secreted molecule. While this question is beyond the scope of our current study, we have revised the manuscript such that we do not overstate our conclusions. This issue is only mentioned a single time (in paragraph 2 of the revised Discussion), and we note that we cannot distinguish between paracrine and autocrine signaling mechanisms.

3. In *cfKO*, one cannot rule out the contribution of epicardial cells as a source of Fstl1. Co-immunostaining with antibodies directed against Fstl1 and *Wt1* would be informative.

Response: Please note that it is reported that the epicardial source of Fstl1 is silenced in the post-MI heart (as shown in Wei et al. *Nature* 2015) and thus would not be a contributor to overall cardiac Fstl1 in the injured heart.

4. In their previous work (Wei et al. *Nature*. 2015), the authors described massive cardiomyocyte-specific Fstl1 staining in the heart 7-14 days post infarction. Why is this not apparent in the present study?

Response: We respectfully disagree with this statement. Wei et al. did not report massive cardiomyocyte specific Fstl1 in the heart. They reported the large upregulation of “myocardial”, not cardiac myocyte, Fstl1 in the post-MI heart, but they did not identify the cell source. As can be seen by the immunofluorescence in Figure 2i from Wei et al., there is little or no overlap of Fstl1 (red) with the cardiac myocyte marker Tnni3 (green). Furthermore, the immunohistochemistry in Figure 2i of Wei et al. appears to us to be very consistent with expression by cardiac fibroblasts, but identification of the cell source was not addressed in their study.

In our current study, we provide multiple lines of evidence to show that fibroblasts are a significant source of cardiac Fstl1 and that the genetic deficiency of Fstl1 in these cells produces a cardiac rupture phenotype.

Fig. 2i

Fig. 2I

5. The physiological data described in Figure 4 and in Table 1 and 2 are convincing. However, the authors focus on the acute phase. Although function is not different in the two genotypes 7 days post MI, it remains possible that *Fstl1* plays a role in maintaining cardiac integrity during the chronic period. Do the authors have investigated cardiac dimensions and function during the chronic phase?

Response: Yes, we assessed cardiac dimension and function in surviving mice of both strains. As can be seen in the Table below, *Fstl1* deficiency in S100a4⁺ cells leads to a statistically significant reduction in fractional shortening at 4 weeks post-MI (A, below). Furthermore, while there was a trend toward greater infarct size at this time point, it was not statistically significant (B, below). These data are also included in the Supplemental data section of the revised manuscript (Fig. EV 4 and Table EV IV).

A. Cardiac dimension and function

	4 weeks after MI		
	WT (n=6)	cfKO (n=5)	P-value
FS (%)	17.28 ± 1.21	11.58 ± 1.25	p=0.0097
LVIDd (mm)	4.58 ± 0.12	4.82 ± 0.33	p=0.4857
LVIDs (mm)	3.78 ± 0.11	4.26 ± 0.34	p=0.3290
IVSd (mm)	0.42 ± 0.06	0.35 ± 0.02	p=0.2670
PWd (mm)	1.15 ± 0.10	0.97 ± 0.09	p=0.2096

FS, fractional shortening; LVIDd, left ventricular internal diastolic dimension; LVIDs, left ventricular internal systolic dimension; IVSd, intraventricular septum diastole; PWd, posterior wall diastole

Unpaired t-test was used for statistical analysis except LVIDs. LVIDs was analyzed by Mann-Whitney test.

Heart samples were harvested at day 28 after MI and processed for paraffin section. The section was stained with Picrosirius red stain to visualize the infarcted area. Size of infarcted area was normalized based upon whole heart size. Error bars represent mean ± SEM (n= 6 and 9 for WT and cfKO, respectively). Statistical analysis was performed by unpaired t-test (two-tailed).

6. Figure EV 4D suggests that *Fstl1* ablation has no impact on fibronectin expression but this is in contrast to what shown in Figure 5B.

Response: Yes; it is little complicated, but this is the crux of the argument. *Fstl1* clearly does not have a direct effect on fibronectin expression (α SMA, etc.) based upon mechanistic cell culture experiments (shown in Fig. EV 4). In contrast, *Fstl1* manipulations lead to changes in fibroblast proliferation and migration in the cell culture experiments (Figures 7 and 8). However, the

deficiency of Fstl1 *in vivo* leads to reductions in fibronectin (α SMA, etc.) expression. Thus, we conclude that the role of Fstl1 in the heart is the early activation of fibroblasts (proliferation and migration) and that the effects of Fstl1-deficiency on myofibroblast differentiation *in vivo* are an indirect, or secondary, effect of having reduced fibroblast activation/number in this context. This model is elaborated in Discussion and shown in Figure 8D. Overall, we strongly believe that our work is of high priority because it represents the most definitive understanding of how Fstl1 contributes to a fibrotic response in any context.

Minor points

1. It would be nice to have control sham hearts for comparison in Figure 1A and B.

Response: A sham is included in these figures, but we acknowledge this is a T=0 time point. However, we point out to the reviewer that we have examined the sham condition at other time points (e.g. T=7 days in Fig 3C) and we see little or no Fstl1 induction.

2. Contrary to what is stated in the text (page 6), Figure 2D does not support expression of Fstl1 in α SMA-positive cells.

Response: We are highly confident of our co-localization data. We include another set of panels for Figure 2D that the reviewer questioned. The modified figure is also shown below.

3. What is the level of S100A expression in *cfKO* post MI (Figure 3)?

Response: Figure 3A was modified to include S100a4 expression in the wild type and *cfKO* hearts. There was a trend toward reduced expression but this was not statistically significant.

4. Data in Figure 5C should be quantified, and normalized per heart section.

Response: The data in Figure 5C was quantified and it is included in the revised manuscript. It is also shown below.

Error bars represent mean \pm SEM. Statistical analysis was performed by Mann-Whitney test ($n=16$ for WT and $n=14$ for cfKO).

5. Data in Figure 7A should be quantified.

Response: Both Figures 7A and B were quantified and they are included in the revised manuscript. They are also shown below.

(A) NRCFBs at passage 1 were stimulated with recombinant Fstl1 (50ng/ml) or vehicle after culture in serum-reduced conditions (FBS 0.5%) for 24 hours. The samples were harvested at the indicated time points after stimulation. The expression of ERK1/2 and tubulin were detected by immunoblotting. Error bars represent mean \pm SEM. Statistical analysis was performed by one-way ANOVA and Tukey's multi comparison test ($n=3$ for each time point). (B) Fstl1 induced phosphorylation of ERK1/2 was ablated by pretreatment with MEK inhibitor PD98059. NRCFBs were treated by PD98059 (5nM) for 30min and then stimulated with recombinant Fstl1 protein (50ng/ml) for 15min. Error bars represent mean \pm SEM. Statistical analysis was performed by one-way ANOVA and Tukey's multi comparison test ($n=3$ for each time point).

Thank you for the submission of your revised manuscript to EMBO Molecular Medicine.

We have now received the enclosed reports from the referees that were asked to re-assess it. As you will see the reviewers are now globally supportive and I am pleased to inform you that we will be able to accept your manuscript pending the following final amendments:

- 1) As you will see, Reviewer 1, does have a few remaining requests for your action, mostly aimed at clarifying a few remaining points. I would like you to carefully consider, and respond to each point, introducing the appropriate textual changes in the manuscript where necessary. I am prepared to make an editorial decision on your next, final version of your manuscript.
- 2) Please note that the callouts in the manuscript to the EV tables are a mix of Arabic and Roman numerals. Please correct to Arabic. Also, there appear to be inconsistent callouts for the main tables, please double check that these are correct and appropriate.
- 3) The EV figures are currently provided as a single ppt file. Please provide separate files for each figure.
- 4) The appendix file is currently a ppt file. Please provide as a pdf file instead.
- 5) Thank you for providing source data files. However, please collect into ONE file per figure (multiple pages to accommodate different figure panels are fine).
- 6) As per our Author Guidelines, the description of all reported data that includes statistical testing must state the name of the statistical test used to generate error bars and P values, the number (n) of independent experiments underlying each data point (not replicate measures of one sample), and the actual P value for each test (not merely 'significant' or ' $P < 0.05$ '). I note that you have provided some, but not all P values.

Please submit your revised manuscript within two weeks. I look forward to seeing a revised form of your manuscript as soon as possible.

***** Reviewer's comments *****

Referee #1 (Remarks):

The authors provided a comprehensive and adequately balanced response to the concerns raised by this reviewer.

An important concern, relating to cardiac fibroblasts as the major source of Fstl1 in the permanent mouse post-MI model, was sufficiently addressed by the authors. Given that S100A4 is not exclusively expressed by cardiac fibroblasts, the authors provided appropriate IF stains demonstrating the absence of Fstl-1 expression in non-cardiomyocyte and non-cardiac fibroblasts, usually present in post-MI myocardium both in the IA and remote zone. This reviewer further agrees that this body of evidence circumvents the need to further demonstrate cardiac rupture rate in other Fstl-1 KO mouse models. As such, the novel data support the authors claim sufficiently that in the model employed (permanent post-MI), cardiac fibroblast might be the predominant but not sole source of secreted Fstl-1.

Another important issue was the difference in Fstl-1 dosage (produced amount post-MI vs. other previously reported ischemic/hypertrophic mouse models), timing and cellular sources. Overall, this reviewer appreciates the arguments brought forward by the authors mostly relating to previously published data on Fstl-1 by themselves and other groups. However, a more suitable approach could have been the demonstration i.e. of a threshold effect on isolated cardiac fibroblasts with respect to proliferation and migration as shown for a single Fstl-1 dose by the authors. The authors should

therefore address this limitation adequately in the revised discussion but are not encouraged anymore to conduct additional experiments. The amount of experiments already done to address the concerns of all three reviewers are extensive enough. This reviewer is of the opinion that critique during a review cycle can be sufficiently answered in many way by the authors and should always be understood as an constructive dialogue. As such, addressing this point in the revised discussion as a potential shortcoming of the study seems to adequate.

The unchanged inflammatory and hypertrophic response in the mouse model employed is surprising but data are data. The authors are encouraged to discuss the differential outcome to other studies with the necessary insight into molecular modes of actions.

Additional biochemical studies (i.e. AMPK and Akt) provide additional inside but do not fully unveil the molecular mode of action of the proposed high dosage effect of Fstl-1 on cardiac fibroblasts. However, it advances the reader's insight beyond initially provided data (i.e. ERK1/2) and opens avenues for continued research on the subject.

Overall, the authors have provided an adequate revised manuscript that significantly improved in quality.

Referee #2 (Remarks):

The authors addressed my points adequately. I have no further comments on this manuscript. I would like to congratulate the authors for the nice piece of work.

Referee #3 (Remarks):

In this revised version of their manuscript, the authors have addressed my concerns. The discussion has been expanded to clarify some of the issues raised by the previous version. I am satisfied with this version of the manuscript.

2nd Revision - authors' response

27 April 2016

Editor:

1) As you will see, Reviewer 1, does have a few remaining requests for your action, mostly aimed at clarifying a few remaining points.

Response: We have carefully considered and responded to Reviewer 1's comments below and have made the appropriate textual changes in the manuscript.

2) Please note that the callouts in the manuscript to the EV tables are a mix of Arabic and Roman numerals. Please correct to Arabic. Also, there appear to be inconsistent callouts for the main tables, please double check that these are correct and appropriate.

Response: We have changed the callouts to the EV tables to Arabic numerals in the manuscript.

3) The EV figures are currently provided as a single ppt file. Please provide separate files for each figure.

Response: EV Figures are now provided as separate ppt files.

4) The appendix file is currently a ppt file. Please provide as a pdf file instead.

Response: The appendix file has been converted to pdf.

5) Thank you for providing source data files. However, please collect into ONE file per figure (multiple pages to accommodate different figure panels are fine).

Response: Source data have been organized into one file per figure.

6) As per our Author Guidelines, the description of all reported data that includes statistical testing must state the name of the statistical test used to generate error bars and P values, the number (n) of independent experiments underlying each data point (not replicate measures of one sample), and the actual P value for each test (not merely 'significant' or ' $P < 0.05$ '). I note that you have provided some, but not all P values.

Response: We have added this information in the text of the manuscript.

The Paper Explained... Please provide a draft summary of your article highlighting:

Response: A draft summary of our article highlighting these points is included on page 33 of the manuscript.

Every published paper now includes a 'Synopsis' to further enhance discoverability. Synopses are displayed on the journal webpage and are freely accessible to all readers. They include a short stand first (maximum of 300 characters, including space) as well as 2-5 one sentence bullet points that summarise the paper. Please write the bullet points to summarise the key NEW findings. They should be designed to be complementary to the abstract - i.e. not repeat the same text. We encourage inclusion of key acronyms and quantitative information (maximum of 30 words / bullet point). Please use the passive voice. Please attach these in a separate file or send them by email, we will incorporate them accordingly.

Response: We have included a Synopsis as a separate file.

You are also welcome to suggest a striking image or visual abstract to illustrate your article. If you do please provide a jpeg file 550 px-wide x 400-px high.

Response: We provide a jpeg file with expanded resolution.

Referee #1 has 2 additional requests:

Another important issue was the difference in Fstl-1 dosage (produced amount post-MI vs. other previously reported ischemic/hypertrophic mouse models), timing and cellular sources.... However, a more suitable approach could have been the demonstration i.e. of a threshold effect on isolated cardiac fibroblasts with respect to proliferation and migration as shown for a single Fstl-1 dose by the authors. The authors should therefore address this limitation adequately in the revised discussion but are not encouraged anymore to conduct additional experiments.

Response: In the revised Discussion we added the text, "It is also possible that quantitative differences in Fstl1 induction levels in the different cardiac injury models have divergent effects on various cardiac cell types that might differ in their threshold responses to this ligand."

The unchanged inflammatory and hypertrophic response in the mouse model employed is surprising but data are data. The authors are encouraged to discuss the differential outcome to other studies with the necessary insight into molecular modes of actions.

Response: In the revised Discussion we added the text, "In other models of cardiac pathology, Fstl1 has been shown to have anti-hypertrophic and anti-inflammatory activities (Ogura et al, 2012, Shimano et al, 2011). However, Fstl1 ablation in fibroblasts did not detectably affect these parameters in the current study, perhaps because the pro-hypertrophic and pro-inflammatory signals in the post-MI heart overwhelm any impact of partial Fstl1 deficiency."

Corresponding Author Name: Kenneth Walsh
 Journal Submitted to: EMBO Molecular Medicine
 Manuscript Number: 06151